# Attribution-Guided and Coverage-Maximized Pruning for Structural MoE Compression

**Yifu Ding** [1 2 3]   **Jiacheng Wang** [1 4]   **Ge Yang** [1 4]   **Yongcheng Jing** [3]
**Jinyang Guo** [1 4]   **Xianglong Liu** [1 2]   **Dacheng Tao** [3]

## Abstract

Mixture-of-Experts (MoE) models scale compute efficiently, yet they remain expensive to deploy due to substantial memory footprint and inference overhead. Prior methods mainly operate at the expert level, either removing whole experts or ranking experts by importance. However, such expert-wise decisions are too coarse to identify redundancy, and often misallocate pruning budgets and limits compression. To alleviate this dilemma, we observe that information in MoE experts is highly concentrated in a few channels, leaving substantial redundancy even in "high importance" experts. Accordingly, we propose a structural pruning framework tailored for MoEs, reforming the prune-ratio objective to maximizing channel-score coverage via an efficient attribution-based approximation. Experiments on DeepSeek and Qwen MoEs retain accuracy under 50% or 25% pruning joinly with 4-bit quantization, reducing the memory footprint of Qwen3-30B-A3B by $5.27\times$, and outperforming state-of-the-art baselines under diverse benchmarks.[1]

## 1. Introduction

Mixture-of-Experts (MoE) architectures have become a dominant paradigm for scaling language models, offering high parameter capacity while maintaining manageable computation by activating only a subset of experts for each token (Xue et al., 2024; Qwen-Team, 2025; Guo et al., 2025a).

This work is completed during Yifu Ding's research attachment at NTU. [1]State Key Laboratory of Complex & Critical Software Environment, Beihang University [2]School of Computer Science and Engineering, Beihang University [3]Nanyang Technological University [4]School of Artificial Intelligence, Beihang University. Correspondence to: Xianglong Liu <xlliu@buaa.edu.cn>.

*Proceedings of the $43^{rd}$ International Conference on Machine Learning*, Seoul, South Korea. PMLR 306, 2026. Copyright 2026 by the author(s).

[1]Our code is available at https://github.com/yifu-ding/MoE-Slimming.

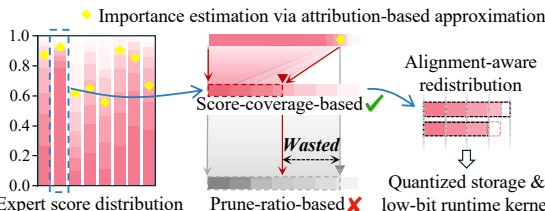

*Figure 1.* Overview of our pruning framework: estimating expert importance via an attribution-based approximation (left), maximizing score coverage to avoid wasting capacity (middle), and applying alignment-aware redistribution for compact storage and kernel-friendly low-bit inference (right).

To effectively deploy large modern MoEs and accelerate the inference, structural pruning, which removes entire channels or experts to yield hardware-efficient dense smaller model, offers a promising solution (Ma et al., 2023; Gao et al., 2024; An et al., 2024; Guo et al., 2024). Quantization, which reduces model bit-widths, is another complementary efficiency approach (Gong et al., 2025; Lv et al., 2026). In contrast to dense models, which use a single FFN per-layer shared by all tokens, MoEs comprise multiple experts with token-dependent routing. Experts are activated at vastly different frequencies and exhibit non-uniform internal redundancy (Huang et al., 2024; Zhang et al., 2024). Consequently, pruning decisions are tightly related to data-dependent activation.

Allocating good prune ratios across heterogeneous experts becomes substantially harder as modern MoEs scale to hundreds of experts, compared to earlier MoEs with only a handful of experts. Expert-wise, loss-based ablations (Zhang et al., 2024; Lu et al., 2024) require evaluating each expert separately, so the cost scales linearly with the number of experts and becomes impractical at scale (Yang et al., 2024; Bai et al., 2025). Routing statistics (He et al., 2025; Lee et al., 2025; Xie et al., 2024) are cheap to collect, but they only capture selection frequency and aggregation proportion, rather than the experts' true contribution. Moreover, both methods make decisions at the expert level, treating each expert as a whole unit and failing to characterize its internal redundancy, namely *how much capacity can be safely removed* under significant expert heterogeneity. As a result, accurate and scalable capacity allocation across experts in

large MoEs remains underexplored.

In this paper, we rethink MoE structural pruning based on our observation that MoE information is highly concentrated in a small fraction of channels, making expert-level importance too coarse to capture internal redundancy. To the best of our knowledge, we are the first to show that even "high-importance" experts may not require large capacity. This motivates a score-coverage-maximized allocation that prioritizes high contributed structures and avoids wasting capacity on low-score tails.

We propose **Attribution-guided and Coverage-Maximized Expert-wise Pruning**, a framework tailored for MoE slimming. As shown in Figure 1, instead of allocating prune ratios directly from expert-level importance, we maximize *channel score coverage* under a global budget, which better aligns with the highly concentrated and unbalanced information distribution in modern MoEs.

Our framework consists of three components, as shown in Figure 1: (1) *Attribution-guided Loss Approximation* (ALA) efficiently estimates expert importance layerwisely, without exhaustive ablation. (2) *Coverage-maximized Budget Allocation* (CBA) uses ALA scores and performs coverage-driven capacity allocation under a global budget, retaining high-contribution channels while pruning low-score tails. (3) *Alignment-Aware Redistribution* (AAR) adjusts dimensions after the initial allocation to satisfy low-bit kernel constraints, ensuring seamless integration with quantized storage and efficient inference.

Our framework achieves impressive results on representative MoE architectures, including DeepSeek and Qwen MoEs, across diverse downstream benchmarks. On general knowledge benchmarks, it delivers over $5\times$ compression with an average accuracy drop of at most 1%. On reasoning benchmarks, the compressed models consistently approach or even surpass the original counterpart across various models and tasks. These results demonstrate the effectiveness of our fine-grained, expert-wise pruning framework and provide a practical path toward efficient MoE deployment.

The main contributions are summarized as follows:

- We observe that MoE information is concentrated in a small fraction of channels, making expert-level importance too coarse to capture expert internal redundancy.

- We are the first to introduce *channel score coverage* as a pruning objective, reformulating capacity allocation as maximizing coverage under a global budget to avoid wasting capacity on low-score tails.

- We propose an *attribution-guided loss approximation* to enable scalable importance expert estimation with $20\times$ fewer GPU hours, and *alignment-aware redistri-*

*bution* for satisfy kernel shape constraints, allowing kernel-friendly storage and efficient inference.

- Experiments on DeepSeek and Qwen MoEs deliver over $5\times$ compression with strong accuracy, with under 1% drop on general knowledge, and 94.5 on MATH500 for Qwen3-30B-A3B under aggressive 50% pruning.

## 2. Related Works

Due to space limitations, a more comprehensive discussion is provided in Appendix Section D.

**MoE Compression.** For efficient deployment of large MoEs, prior work explores: (i) *Expert trimming* and *expert skipping* to reduce runtime computation (Liu et al., 2024a; Bai et al., 2025; Lu et al., 2024; Chen et al., 2025a; Huang et al., 2025). (ii) *Expert slimming* to compress each expert via pruning, quantization, or low-rank factorization (Yang et al., 2024; Xie et al., 2024; Chen et al., 2025b; Guo et al., 2024; Chen et al., 2024). Concurrent to this work, an anonymized submission (provided in the supplementary material) studies MoE pruning with a focus on structural pruning along the hidden dimension (Anonymous, 2026). (iii) *Expert merging* to combine similar experts (Zhao et al., 2025; Guo et al., 2025b). Most approaches operate at the granularity of whole experts or apply uniform compression for each expert, while only limited work explores heterogeneous compression across experts, e.g., different low-rank ranks (Yang et al., 2024) and mixed-precision bitwidth assignments (Chen et al., 2025b)).

**Expert Importance Estimation.** A key challenge in MoE compression is estimating per-expert importance. Existing approaches commonly rely on router outputs (gate weights, token usage) (He et al., 2025; Lee et al., 2025; Huang et al., 2024), activation-based metrics (Dong et al., 2025; Zhao et al., 2025), performance-based criteria (e.g., loss or accuracy degradation under ablation) (Liu et al., 2024a; Yang et al., 2024), or learnable scalars (Bai et al., 2025). However, these signals are often inadequate for MoE slimming because they operate at the expert level and ignore expert internal information concentration, making them only suitable for expert trimming rather than fine-grained expert slimming. Our approach advances prior works by replacing costly expert-wise ablation with efficient approximation, and further goes beyond ranking experts to channel-level budgets allocation via global score-coverage maximization.

## 3. Pre-analysis: *The Inherent Difficulty of Expert-Level Importance Estimation*

MoEs sparsely route tokens across experts, and experts contribute unequally to final performance, making expert importance estimation a key problem in MoE compression.

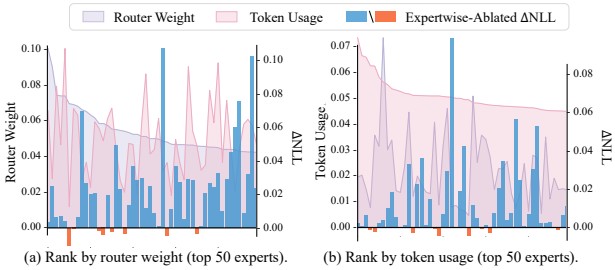

(a) Rank by router weight (top 50 experts).     (b) Rank by token usage (top 50 experts).

*Figure 2.* Misalignment between router outputs and expert-wise ablated NLL. (a) and (b) rank the top 50 experts by router weight and token usage. The NLL (bars) demonstrates a weak correlation with router outputs. Notably, the orange bars highlight that even selected experts can provide negative contributions.

However, existing methods typically rely on router outputs or expert statistics, which are often coarse and unreliable for fine-grained slimming allocations. Below, we revisit common metrics and their limitations, and motivate our approach by highlighting a fundamental mismatch between expert contribution and internal redundancy.

### 3.1. Limitation of Heuristic Metrics

Existing metrics suffer from two key limitations: (1) **router outputs** (e.g., routing weights or token usage) only quantify token engagement but do not indicate whether an expert's output is beneficial or harmful; (2) **raw statistics** (e.g., weight, activation or gradients) exhibit layer-dependent magnitude across layers, and also poorly correlate with the actual contribution of experts within a layer.

**Routers can have wrong choices.** Some prior works estimate expert importance using router outputs, such as post-softmax probabilities or the number of tokens routed to each expert. However, Figure 2 shows that these routing statistics can be seriously misaligned with an expert's true contribution, measured by expert-wise ablated Negative Log-Likelihood (NLL). Concretely, we plot expert-wise ablated $\Delta$NLL (bars) alongside router probabilities and token usage on Qwen1.5-MoE-A2.7B: (a) plots the top 50 experts ranked by router weight and (b) ranked experts by token usage. Empirically, both router probabilities and token usage exhibit weak correlation with $\Delta$NLL. Highly prioritized or frequently activated experts can cause only a minor loss increase when removed (blue bars), and some even reduce the loss (orange bars below zero). This suggests that routing signals mainly reflect selection and how the experts' output are aggregated, rather than whether an expert is beneficial, and the selected expert can be noisy or even harmful.

> **Takeaway 1:** Router-derived statistics (post-softmax weights, token usage) only reflect the engagement of experts, but do not tell the actual contribution of experts.

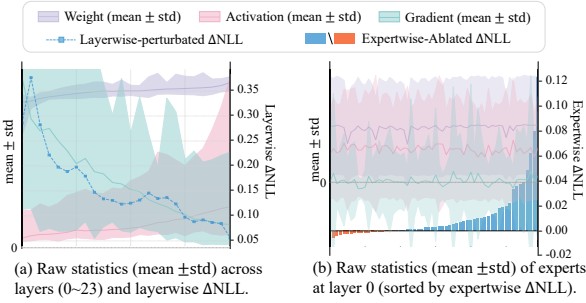

(a) Raw statistics (mean ±std) across layers (0~23) and layerwise $\Delta$NLL.    (b) Raw statistics (mean ±std) of experts at layer 0 (sorted by expertwise $\Delta$NLL).

*Figure 3.* Incomparability of raw statistics (weight, activation and gradient) across layers and experts. (a) shows raw statistics of weights and activations grow monotonically while gradients decay with depth; and (b) reveals intra-layer uncorrelation between these statistics with actual $\Delta$NLL when ablated.

**Incomparability of raw statistics across layers or experts.** Beyond router signals, another common heuristic estimate expert importance from raw forward or backward statistics (e.g., weights, activations, or gradients). However, these quantities are not comparable across layers and often not informative across experts within a layer, making them unreliable as direct importance proxies. (1) *Cross-layer magnitude bias.* In Figure 3(a), the layer-wise mean±std of raw weights and activations exhibits a depth-dependent trend, whereas gradients decay with depth. Such behavior arises due to residual accumulation, normalization, and so on. In contrast, layerwise $\Delta$NLL (blue markers) follows a different pattern and does not align with any raw statistic, indicating that magnitudes are inherently layer-dependent and unsuitable for cross-layer comparison. (2) *Intra-Layer non-correlation.* Figure 3(b) shows a similar issue within a single layer: after sorting experts by expert-wise ablated $\Delta$NLL (bars), the corresponding weight, activation, and gradient statistics (mean±std) exhibit no meaningful relation to loss impact, failing to distinguish helpful experts.

> **Takeaway 2:** Raw statistics (weights, activations, gradients) exhibit weak cross-layer and intra-layer correlation with actual loss when removing the expert, fail to reliably represent the expert importance.

### 3.2. Mismatch between Redundancy and Contribution

Some prior work estimates expert importance by measuring the loss increase when an expert is entirely removed. While this yields an expert-wise ranking, it does not indicate how much capacity can be safely removed within each expert.

**Visualization of channel redundancy.** To examine how information is distributed inside each expert, in Figure 4(a), we sort channels by their scores (see Appendix Section C.3.1) in descending order, and plot the cumulative fraction of the score covered by the top-$k$% channels. The results reveal pronounced heterogeneity in intra-expert redundancy: for some experts, nearly 40% channels conveys

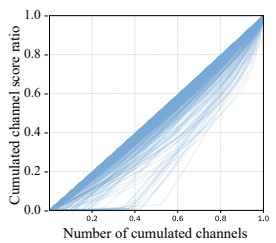
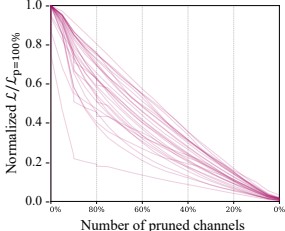

(a) Cumulative channel score curves. (b) Layerwise loss under various prune ratio.

*Figure 4.* (a) Cumulative channel score distribution, which reveal that many experts possess highly centralized channels. (b) Layer-wise output loss under various prune ratio, for some experts the loss drops rapidly after keeping only a small fraction of channels.

negligible information, whereas other experts exhibit much weaker concentration. This expert-specific redundancy cannot be captured by expert-level importance signals alone.

**Contribution can be recovered by few channels.** Given the concentration patterns, whole-expert ablation becomes an overly coarse proxy for redundancy. Figure 4(b) shows that even when removing all channels causes a noticeable $\Delta$NLL, the loss may drop rapidly as only a small fraction of channels is restored. Therefore, whole-expert ablation mainly serve as a binary signal about whether an expert is important, rather than quantifying how much redundancy exists within the expert or how pruning budgets should be allocated across channels.

> **Takeaway 3:** Expert-wise ablation measures expert-level contribution ($\Delta$NLL), but do not reflect the internal redundancy: channel scores can be highly concentrated in a small fraction of channels.

In conclusion, there is still no accurate and scalable metric that quantifies such redundancy across layers and experts, which motivates a fine-grained pruning strategy to determine the actual capacity of each expert.

## 4. Proposed Method

In this section, we present a structural-pruning method for MoE slimming. An overview is shown in Figure 5.

### 4.1. Attribution-Guided Loss Approximation

**Rationale.** As Section 3.1 show, routing outputs and raw statistics can be weakly aligned with an expert's true impact on the final output. This raises a key question: *how can we obtain an accurate yet efficient proxy of loss impact for expert-wise prune budget allocation?*

In this subsection, we propose an Attribution-based Loss Approximation (ALA) to estimate expert contributions, producing a scalable expert-wise loss proxy that initializes our coverage-maximized pruning algorithm in Section 4.2.

**Derivation.** Let $h_\ell \in \mathbb{R}^d$ be the input hidden state of the MoE block in layer $\ell$, and the output can be written as

$$y_\ell = \sum_{e \in \mathcal{E}_\ell} g_{\ell,e}(h_\ell) \, z_{\ell,e}. \tag{1}$$

where $z_{\ell,e} = f_{\ell,e}(h_\ell)$ and $g_{\ell,e}(h_\ell) \geq 0$ are the expert output and the router probability for selected experts $e \in \mathcal{E}_\ell$. Removing expert $e$ corresponds to setting $z_{\ell,e} = 0$, which perturbs the layer output by

$$\Delta y_\ell^{(e)} = -g_{\ell,e} \, z_{\ell,e}. \tag{2}$$

Let $\mathcal{L}$ denote the loss evaluated against the original layer output. We approximate the loss change using a first-order Taylor expansion around $y_\ell$. Thus, the loss change induced by removing expert $e$ is

$$\Delta \mathcal{L}^{(e)} \approx \left( \frac{\partial \mathcal{L}}{\partial y_\ell} \right)^\top \Delta y_\ell^{(e)} = - \left( \frac{\partial \mathcal{L}}{\partial y_\ell} \right)^\top (g_{\ell,e} z_{\ell,e}). \tag{3}$$

Using the chain rule, the loss gradient with respect to the expert output satisfies $\frac{\partial \mathcal{L}}{\partial z_{\ell,e}} = g_{\ell,e} \frac{\partial \mathcal{L}}{\partial y_\ell}$. Put this into the first-order expansion yields the final approximation form

$$\Delta \mathcal{L}_\ell^{(e)} \approx - \left( \frac{\partial \mathcal{L}_\ell}{\partial z_{\ell,e}} \right)^\top z_{\ell,e}, \tag{4}$$

which serves as a proxy of expert contribution, and we compute for all experts within a layer in one backward pass.

**Implementation and efficiency.** We collect $\Delta \mathcal{L}_\ell^{(e)}$ on a calibration set of roughly 3M tokens using an exponential moving average (EMA). We perturb all experts at layer $\ell$ by uniformly scaling their activation outputs with a small factor, and then apply a simple square-root smoothing to the loss, obtaining the expert-wise importance prior $\phi$.

*Table 1.* Comparisons of time costs (GPU hours) between loss-based importance estimation using expert-wise ablation and ours.

| Model | Layers | Experts | Expert-ablated (h) | Ours (h) |
|---|---|---|---|---|
| DeepSeek-MoE-16B | 28 | 64 | 23.67 | 1.70 |
| DeepSeek-V2-Lite | 27 | 64 | 20.05 | 1.43 |
| Qwen1.5-MoE-A2.7B | 24 | 60 | 21.77 | 1.54 |
| Qwen3-30B-A3B | 48 | 128 | 130.10 | 5.23 |

We compare calibration time in Table 1 against expert-wise ablation under the same data amount and iterations. Our method reduces 14-26× time cost due to a smaller search space. While heuristics such as greedy search (Cao et al., 2024) or genetic algorithms (Liu et al., 2024a) can reduce ablation cost, they can probably fall into local optimum. Meanwhile, it is noticeable that none of them has been validated on MoEs with hundreds of experts.

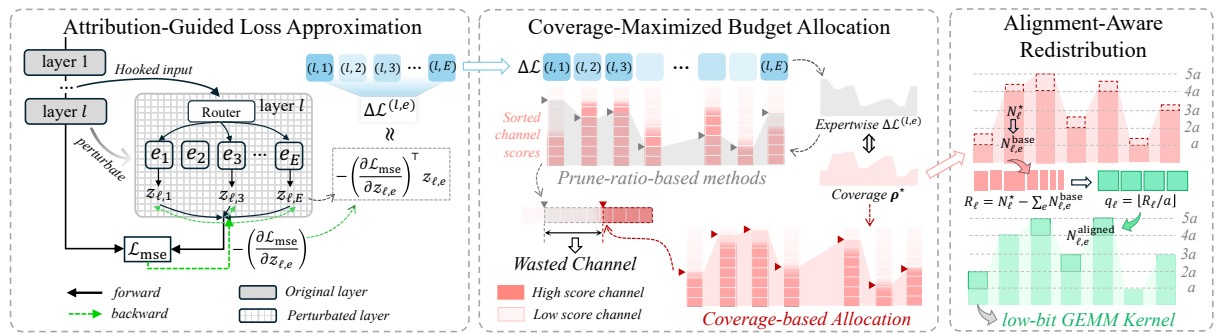

*Figure 5.* The overview of Attribution-Guided & Coverage-Maximized Expert-wise Pruning framework for MoE models.

---

**Algorithm 1** Coverage-Maximized Allocation Search

1: **Input:** Score allocation weights $\phi \in \mathbb{R}_+^{|\mathcal{G}|}$; prefix sums $\{\mathcal{S}_g(n)\}_{g\in\mathcal{G}}$; total scores $\{\mathrm{S}_g^{tot}\}_{g\in\mathcal{G}}$; channel budget $N_{\text{budget}}$; total channels $N^{tot}$; tolerance $\varepsilon$.
2: **Output:** Channel budgets $\{N_g^\star\}_{g\in\mathcal{G}}$
3: $\quad \alpha_{\min} \leftarrow 0, \alpha_{\max} \leftarrow 1$
4: **while** $\alpha_{\min} < \alpha_{\max}$ **do**
5: $\quad \alpha \leftarrow (\alpha_{\min} + \alpha_{\max})/2$
6: $\quad \boldsymbol{\rho} \leftarrow \min(\alpha\,\boldsymbol{\phi}, 1)$
7: $\quad N(\boldsymbol{\rho}) \leftarrow \sum_{g\in\mathcal{G}} \min\{n \mid \mathcal{S}_g(n) \geq \rho_g(\alpha)\,\mathrm{S}_g^{tot}\}$
8: $\quad$ **if** $|N(\boldsymbol{\rho}) - N_{\text{budget}}| \leq \varepsilon\,N^{tot}$ **then**
9: $\quad\quad N_g^\star \leftarrow \min\{n \mid \mathcal{S}_g(n) \geq \rho_g(\alpha)\,\mathrm{S}_g^{tot}\}, \forall g \in \mathcal{G}$
10: $\quad\quad$ **break**
11: $\quad$ **end if**
12: $\quad$ **if** $N(\boldsymbol{\rho}(\alpha)) > N_{\text{budget}}$ **then**
13: $\quad\quad \alpha_{\max} \leftarrow \alpha$
14: $\quad$ **else**
15: $\quad\quad \alpha_{\min} \leftarrow \alpha$
16: $\quad$ **end if**
17: **end while**
18: **return** $\{N_g^\star\}_{g\in\mathcal{G}}$

---

### 4.2. Coverage-Maximized Budget Allocation

Motivated by the mismatch between experts contribution and their internal redundancy observed in Section 3.2, we propose a new objective that directly rewards retaining the concentrated, high-contribution channels.

**Unified coverage formulation for inter- and intra-layer allocation.** Consider a group $\mathcal{G}$, which can be either all layers or all experts within a single layer. Each layer or expert $g \in \mathcal{G}$ contains channels $c \in \mathcal{C}_g$ with non-negative scores $s_{g,c} \geq 0$. Sorting channels by $s_{g,c}$ in descending order, then we notate them as $s_{g,(1)} \geq \cdots \geq s_{g,(|\mathcal{C}_g|)}$. And then, we precompute the prefix sums of $n$ channels as

$$\mathcal{S}_g(n) = \sum_{i=1}^{n} s_{g,(i)}, \qquad \mathrm{S}_g^{tot} = \mathcal{S}_g(|\mathcal{C}_g|), \qquad (5)$$

Given precomputed prefix sums, the coverage ratio for top-$n$ channels is computed directly as $\rho_g(n) = \mathcal{S}_g(n)/\mathrm{S}_g^{tot}$.

The core idea of our algorithm is to change the objective of *prune ratio* allocation to the *channel score coverage* allocation. Given a global prune target $p$, we have total channel budget $N_{\text{budget}}(p) = (1-p)N^{tot} = (1-p)\sum_{g\in\mathcal{G}}|\mathcal{C}_g|$. We allocate channels by searching for the largest target coverage vector $\boldsymbol{\rho} \in [0,1]^{|\mathcal{G}|}$ that maximizes total covered score while retaining the minimum number of channels:

$$N(\boldsymbol{\rho}) = \sum_{g\in\mathcal{G}} N_g(\rho_g) = \sum_{g\in\mathcal{G}} \min\{n \mid \mathcal{S}_g(n) \geq \rho_g\,\mathrm{S}_g^{tot}\}, \quad (6)$$

where each $\rho_g \in \boldsymbol{\rho}$ corresponds to the coverage ratio of $g$, $N(\boldsymbol{\rho})$ is the minimal number of channels needed to reach coverage $\boldsymbol{\rho}$. Since $\mathcal{S}_g(n)$ is monotone in $n$, $N(\boldsymbol{\rho})$ can be obtained efficiently via binary search (Appendix Algorithm 3).

The pipeline of CBA is shown in Algorithm 1. We initialize $\boldsymbol{\rho}$ using non-negative importance prior $\boldsymbol{\phi} \in \mathbb{R}^{|\mathcal{G}|}$ derived from ALA, and a single scaling factor $\alpha$ (line 6 in Algorithm 1). We apply binary search over $\alpha$ to find the largest $\alpha^\star$ such that $N(\boldsymbol{\rho}(\alpha^\star)) \leq N_{\text{budget}}$, which yields the final budgets for each item $g \in \mathcal{G}$: $N_g^\star = N_g(\rho_g(\alpha^\star))$.

**Inter-layer vs. intra-layer instantiation.** The procedure above is identical for inter-layer and intra-layer allocation, which only differ in the definition of group $\mathcal{G}$ and the initialization of importance estimation $\boldsymbol{\phi}$: (1) *Inter-layer allocation.* $\mathcal{G}$ consists of all layers, and $\mathcal{C}_g$ ($\forall g \in \mathcal{G}$) includes all channels in one layer. We set $\boldsymbol{\phi}$ using the layerwise loss, and our algorithm producing budgets $N_\ell^\star$ for all layers. (2) *Intra-layer allocation.* $\mathcal{G} = \{(\ell, 1), \ldots, (\ell, E)\}$ contains all experts at layer $\ell$, and $\mathcal{C}_g$ means channels for expert $g \in \mathcal{G}$. $\boldsymbol{\phi}_\ell$ is derived from our ALA (Section 4.1), and run the same search under the layer budget $N_\ell^\star$ to obtain $N_{\ell,e}^\star, \forall e \in \mathcal{E}_\ell$.

Overall, our CBA algorithm takes the ALA outcome as budget initialization, and translates them into kept channels by maximizing the accumulated scores within each expert. As illustrated in Figure 5, unlike prune-by-ratio baselines (gray) that can waste capacity on low-score tails, our method retains only high-score channels to maximize score coverage (red). Time breakdown of CBA is provided in Appendix Table 12, and more details are in Appendix Section A.1.

### 4.3. Alignment-Aware Redistribution

**Rationale.** To remain compatible with low-bit quantization after pruning, inference backends (e.g., BitsAndBytes) require the input dimensions to be multiples of a hardware-friendly block size. Otherwise, frameworks may (i) trigger warnings and fall back to slower generic implementations, (ii) suffer degraded throughput. For example, Qwen3-30B-A3B drops from 14.21 tokens/s when dimensions are aligned to 128, while 10.23 tokens/s when not aligned, see Table 2. And (iii) incur padding that wastes storage and compute while conveying no information. Qwen3-30B-A3B would have 4.1% padded channels, corresponding to $\approx 4.0 \times 10^8$ wasted parameters (see Appendix Section B.2).

*Table 2.* Throughput and latency of Qwen MoE models with and without channel alignment (under 50% sparsity).

| Model | Alignment ($a$) | Tokens/s | Latency (ms) |
|---|---|---|---|
| Qwen1.5-MoE-A2.7B | – | 31.41 | 31.84 |
|  | 64 | 33.92 | 29.48 |
|  | 128 | 37.12 | 26.94 |
| Qwen3-30B-A3B | – | 10.23 | 97.75 |
|  | 64 | 12.98 | 77.03 |
|  | 128 | 14.21 | 70.38 |

Our coverage-based allocation produces per-expert channel budgets $N_{\ell,e}$ optimal for score coverage, but may violate low-bit GEMM constraints that require channel dimensions to be multiples of a block size $a$ (e.g., 64 or 128). We therefore apply an Alignment-Aware Redistribution (AAR) that converts $\{N_{\ell,e}\}$ into aligned budgets $\{N_{\ell,e}^{\text{aligned}}\}$ while approaching as close as possible to the original allocation.

**Downward alignment.** First, we drop extremely small experts by a minimal channel threshold $m$: experts with $N_{\ell,e} < m$ are set to zero and excluded from redistribution. Because overly slim experts can convey more noise than information. For each remaining expert, we apply downward alignment to the nearest multiple of $a$, producing a kernel-compatible base budget by $N_{\ell,e}^{\text{base}} = \lfloor \tilde{N}_{\ell,e}/a \rfloor \cdot a$. The released quota after rounding in layer $\ell$ is $R_\ell = N_\ell^\star - \sum_e N_{\ell,e}^{\text{base}}$ (red slices), corresponding to $q_\ell = \lfloor R_\ell/a \rfloor$ additional $a$-blocks that can be reassigned.

**Hamilton largest-remainder apportionment.** We perform alignment-aware redistribution via Hamilton's largest-remainder rule. After rounding each active expert's channel budget down to the nearest multiple of $a$, we obtain per-expert fractional remainders $r_{\ell,e} \in [0,1)$ that quantify how close each expert is to the next $a$-block. We then allocate the $q_\ell$ available $a$-blocks to the experts with the largest remainders, which can be written as

$$b_{\ell,e} = \mathbb{I}\left[e \in \{\pi(1), \ldots, \pi(q_\ell)\}\right], \quad (7)$$

where $\pi$ sorts experts in descending order of $r_{\ell,e}$. The final aligned channel budgets are

$$N_{\ell,e}' = N_{\ell,e}^{\text{base}} + a \cdot b_{\ell,e}. \quad (8)$$

This preserves the coverage-based allocation as closely as possible while ensuring divisibility by $a$ for efficient low-bit kernels. The complete redistribution procedure and all implementation details are provided in Appendix Section A.2.

## 5. Experiments

### 5.1. Experimental Setup

**Models and Compared Methods.** We evaluate our method on representative open-source MoEs covering different scales, including DeepSeek-MoE-16B (Dai et al., 2024), DeepSeek-V2-Lite (DeepSeek-AI, 2024), Qwen1.5-MoE-A2.7B (Team, 2024), and Qwen3-30B-A3B-Thinking (Qwen-Team, 2025). We compare against recent LLM or MoE compression methods, including Wanda (Sun et al., 2023) using unstructural pruning, MoNE (Zhang et al., 2025)) with structural pruning, both of which denoted as $\mathbf{P}_{\text{x\%}}$. EAC-MoE (Chen et al., 2025a) and He et al. jointly combine pruning and quantization ($\mathbf{P}_{\text{x\%}}$ $\mathbf{Q}_{\text{yb}}$). MoE-I$^2$ (Yang et al., 2024) proposes low-rank decomposition ($\mathbf{L}_{\text{x\%}}$), and PuzzleMoE (Zhao et al., 2025) applies expert merge ($\mathbf{M}_{\text{x\%}}$). Here, x% represents the parameter reduction ratio, and yb means it uses y-bit quantization. Detailed introductions can be found in Section C.1.

**Implementations.** We adopt 25% channel pruning with 4-bit quantized via alignment, noted as Ours$_Q$ ($\mathbf{P}_{\text{25\%}}$ $\mathbf{Q}_{\text{4b}}$), and more aggressive 50% pruning without quantization or alignment, noted as Ours ($\mathbf{P}_{\text{50\%}}$). We generate pruning allocation using C4 (Raffel et al., 2019) for general benchmarks for knowledge, GSM8K (Cobbe et al., 2021) or OpenCodeReasoning (Ahmad et al., 2025) for math and code, followed by lightweight fine-tuning on Alpaca (Taori et al., 2023). Extended configurations are provided in Appendix Section C.1.

### 5.2. Overall Results

**Results on General Tasks.** Table 3 and Table 5 report zero-shot accuracy on knowledge tasks together with storage with Qwen MoEs and DeepSeek MoEs, respectively. Under the quantization-aware setting, Ours$_Q$ consistently preserves or improves accuracy while substantially reducing storage across all models. In particular, on Deepseek-MoE-16B, Qwen1.5-MoE-A2.7B and Qwen3-30B-A3B, Ours$_Q$ even surpasses the original model after lightweigt fine-tuning on average performance, with more than $5\times$ storage reduction by jointly using structural pruning and quantization. On Deepseek-V2-Lite, Ours$_Q$ also achieves better performance even under more aggressive compres-

*Table 3.* Comparison on Qwen MoE models. MMLU is evaluated under 5-shot setting, while other tasks are evaluated zero-shot.

| Method | Type | Storage | ARC-c | ARC-e | Hella | PiQA | BoolQ | Wino | MMLU | Avg |
|---|---|---|---|---|---|---|---|---|---|---|
| | | | | | **Qwen1.5-MoE-A2.7B** | | | | | |
| Baseline | – | 28.67GB | 40.41 | 69.44 | 77.17 | 80.79 | 79.57 | 69.77 | 61.08 | 68.32 |
| Wanda | $P_{25\%}$ | 28.67GB | 40.70 | 69.07 | 76.48 | 79.87 | 79.54 | 69.14 | 57.55 | 67.48 |
| Wanda | $P_{50\%}$ | 28.67GB | 38.48 | 65.99 | 71.92 | 78.07 | 76.57 | 66.61 | 53.82 | 64.49 |
| Wanda | $P_{25\%}$ $Q_{4b}$ | 7.10GB | 41.64 | 69.74 | 76.48 | 79.38 | 79.36 | 69.22 | 60.34 | 68.02 |
| MoNE | $P_{25\%}$ | 22.40GB | 42.15 | 75.34 | 54.47 | 77.47 | 74.86 | 72.10 | 56.19 | 64.65 |
| MoNE | $P_{50\%}$ | 16.17GB | 32.17 | 64.81 | 43.52 | 70.13 | 69.05 | 63.61 | 45.67 | 55.57 |
| MoNE | $P_{25\%}$ $Q_{4b}$ | 5.60GB | 40.36 | 59.10 | 63.06 | 81.44 | 69.37 | 66.49 | 55.39 | 62.17 |
| He et al. | $P_{50\%}$ | 16.17GB | 26.96 | 42.21 | 45.46 | 68.23 | 63.55 | 53.35 | 31.52 | 47.33 |
| He et al. | $P_{25\%}$ $Q_{4b}$ | 5.60GB | 38.91 | 60.69 | 71.22 | 77.91 | 68.41 | 63.85 | 52.73 | 61.96 |
| EAC-MoE | $P_{38\%}$ $Q_{3.03b}$ | 4.35GB | 41.30 | 66.71 | 74.97 | 79.16 | 75.35 | 68.75 | 55.96 | 61.77 |
| MoE-I$^2$ | $L_{53.98\%}$ | 15.18GB | 41.13 | 71.68 | 53.08 | – | 75.08 | 66.54 | – | 57.82 |
| PuzzleMoE | $M_{25\%}$ | 22.40GB | 40.9 | 73.4 | 57.3 | 79.7 | 79.2 | 69.6 | 60.4 | 65.8 |
| PuzzleMoE | $M_{50\%}$ | 16.17GB | 40.7 | 73.5 | 56.5 | 79.4 | 78.6 | 69.4 | 60.0 | 65.4 |
| C-Prune | $P_{25\%}$ $Q_{4b}$ | 5.60GB | 40.00 | 62.70 | 63.06 | 78.92 | 77.12 | 67.75 | 56.68 | 63.75 |
| **Ours** | $P_{50\%}$ | **16.17GB** | **40.53** | **70.45** | **73.11** | **77.04** | **78.93** | **65.27** | **50.67** | **65.14** |
| **Ours**$_Q$ | $P_{25\%}$ $Q_{4b}$ | **5.60GB** | **45.13** | **73.19** | **75.01** | **79.00** | **84.22** | **68.67** | **58.29** | **69.07** |
| | | | | | **Qwen3-MoE-30B-A3B** | | | | | |
| Baseline | – | 61.06GB | 52.70 | 79.30 | 78.50 | 79.60 | 88.70 | 72.85 | 77.80 | 75.64 |
| Wanda | $P_{25\%}$ | 61.06GB | 26.45 | 33.71 | 25.43 | 52.12 | 38.50 | 51.14 | 22.95 | 35.76 |
| Wanda | $P_{50\%}$ | 61.06GB | 24.57 | 27.23 | 26.48 | 53.75 | 40.55 | 50.12 | 22.93 | 35.09 |
| PuzzleMoE | $M_{25\%}$ | 46.56GB | 51.6 | 78.9 | 58.3 | 79.3 | 88.2 | 70.4 | 76.6 | 71.9 |
| PuzzleMoE | $M_{50\%}$ | 32.07GB | 51.0 | 78.5 | 57.1 | 78.9 | 88.0 | 70.1 | 75.1 | 71.2 |
| **Ours** | $P_{50\%}$ | **32.07GB** | **52.13** | **78.03** | **77.19** | **79.33** | **83.36** | **71.59** | **61.27** | **70.81** |
| **Ours**$_Q$ | $P_{25\%}$ $Q_{4b}$ | **11.64GB** | **57.94** | **81.94** | **78.87** | **81.18** | **84.22** | **72.45** | **73.04** | **75.66** |

*Table 4.* Reasoning benchmarks with math and code tasks.

| Method | Type | GSM8K | HumanEval | MBPP |
|---|---|---|---|---|
| | | **Qwen1.5-MoE-A2.7B** | | |
| Baseline | – | 61.5 | 34.2 | 36.6 |
| C-Prune | $P_{20\%}$ | 39.4 | 32.9 | – |
| **Ours** | $P_{50\%}$ | **58.2** | **38.1** | **36.6** |
| **Ours**$_Q$ | $P_{25\%}$ $Q_{4b}$ | **58.5** | **36.6** | **38.5** |
| | | **DeepSeek-V2-Lite** | | |
| Baseline | – | 30.9 | 32.3 | 43.2 |
| C-Prune | $P_{20\%}$ | 26.4 | 18.9 | – |
| **Ours** | $P_{50\%}$ | **33.7** | **28.7** | **44.4** |
| **Ours**$_Q$ | $P_{25\%}$ $Q_{4b}$ | **29.3** | **23.8** | **37.4** |

| Method | Type | MATH500 | AIME25 | LCB |
|---|---|---|---|---|
| | | **Qwen3-MoE-30B-A3B** | | |
| Baseline | – | 92.8 | 76.7 | 62.6 |
| **Ours** | $P_{50\%}$ | **95.0** | **64.0** | **–** |
| **Ours**$_Q$ | $P_{25\%}$ $Q_{4b}$ | **94.5** | **75.0** | **51.0** |

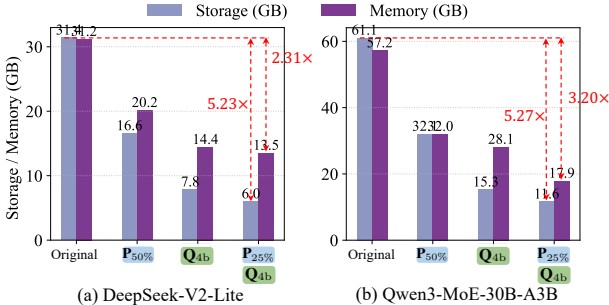

*Figure 6.* Comparison of storage and runtime memory usage (GB).

sion ratio compared to Wanda and MoNE. Overall, results show that our attribution-guided, coverage-maximized allocation achieves strong compression with negligible accuracy loss, and alignment-aware redistribution allows us to integrate with low-bit quantization to achieve further storage saving. We further compare channel-level pruning with expert-level pruning baselines at matched storage budgets in Appendix Section C.2.1, where the Pareto frontier in Figure 9 shows that the advantage becomes larger as the compression budget tightens.

**Results on Reasoning Benchmarks.** We also report accuracy and pass@1 performance on math and code reasoning

tasks in Table 4. On Qwen1.5-MoE-A2.7B, our method in both quantization and non-quantization settings largely retains GSM8K accuracy (58.20 vs. 61.50) while improving HumanEval from 34.20 to 38.14. In contrast, previous method (Guo et al., 2025b) which also allocate different prune ratio on experts based on similarity clustering degrades reasoning accuracy, especially on GSM8K. For the larger Qwen3-MoE-30B-A3B, our method remains robust at higher difficulty, which reaches 95.0 on MATH500 under 50% sparsity, indicating that attribution-guided coverage allocation can preserve the critical intermediate representation space while reduce noisy informations during complex reasoning even under aggressive structural compression.

**Storage and Memory Reduction.** We report the storage footprint and peak memory usage during runtime in Figure 6. Our method yields substantial memory savings. Applying $P_{50\%}$ nearly halves peak memory on all MoEs, e.g.,

*Table 5.* Comparison on Deepseek MoE models. MMLU is evaluated under 5-shot setting, while other tasks are evaluated zero-shot.

| Method | Type | Storage | ARC-c | ARC-e | Hella | PiQA | BoolQ | Wino | MMLU | Avg |
|---|---|---|---|---|---|---|---|---|---|---|
| | | | Deepseek-MoE-16B | | | | | | | |
| Baseline | – | 32.7GB | 47.53 | 73.19 | 77.43 | 80.52 | 72.57 | 69.93 | 38.18 | 65.62 |
| Wanda | $P_{25\%}$ | 32.7GB | 44.54 | 72.26 | 57.58 | 78.51 | 74.16 | 70.24 | 33.63 | 61.56 |
| Wanda | $P_{50\%}$ | 32.7GB | 44.37 | 71.46 | 54.09 | 77.86 | 75.11 | 69.85 | 32.99 | 60.82 |
| MoNE | $P_{25\%}$ | 25.04GB | 22.70 | 25.08 | 25.89 | 53.37 | 37.83 | 49.57 | 23.12 | 33.94 |
| MoNE | $P_{50\%}$ | 17.34GB | 24.53 | 24.58 | 25.57 | 49.51 | 37.83 | 49.57 | 23.12 | 33.24 |
| He et al. | $P_{50\%}$ | 17.34GB | 28.67 | 41.41 | 53.67 | 68.17 | 38.65 | 55.17 | 23.47 | 44.17 |
| He et al. | $P_{25\%}$ $Q_{4b}$ | 7.70GB | 44.00 | 70.75 | 74.50 | 78.50 | 66.00 | 67.30 | 27.90 | 59.70 |
| EAC-MoE | $P_{11\%}$ $Q_{3.03b}$ | 7.19GB | 46.16 | 73.15 | 75.55 | 79.54 | 75.02 | 70.24 | 37.45 | 65.30 |
| EAC-MoE | $P_{38\%}$ $Q_{3.03b}$ | 4.47GB | 45.05 | 71.55 | 72.86 | 77.20 | 72.51 | 66.46 | 33.33 | 62.70 |
| PuzzleMoE | $M_{25\%}$ | 25.04GB | 44.0 | 75.7 | 57.2 | 78.7 | 73.1 | 70.6 | 37.2 | 62.36 |
| PuzzleMoE | $M_{50\%}$ | 17.34GB | 43.0 | 75.2 | 56.3 | 78.4 | 74.5 | 70.3 | 36.9 | 62.08 |
| **Ours** | $P_{50\%}$ | **17.34GB** | **42.58** | **71.38** | **70.50** | **77.97** | **68.78** | **68.75** | **31.66** | **62.34** |
| **Ours$_Q$** | $P_{25\%}$ $Q_{4b}$ | **6.26GB** | **43.77** | **73.74** | **73.77** | **79.60** | **73.55** | **71.11** | **35.70** | **63.78** |
| | | | DeepSeek-V2-Lite | | | | | | | |
| Baseline | – | 31.41GB | 46.93 | 78.37 | 77.98 | 80.20 | 79.82 | 71.35 | 45.58 | 68.60 |
| Wanda | $P_{25\%}$ | 31.41GB | 46.84 | 76.64 | 58.44 | 79.65 | 79.39 | 71.51 | 53.84 | 66.62 |
| Wanda | $P_{50\%}$ | 31.41GB | 31.40 | 50.84 | 35.41 | 64.15 | 63.18 | 59.27 | 40.81 | 49.29 |
| Wanda | $P_{25\%}$ $Q_{4b}$ | 7.70GB | 46.59 | 76.64 | 77.12 | 79.38 | 78.99 | 71.74 | 53.94 | 69.20 |
| MoNE | $P_{25\%}$ | 24.00GB | 46.67 | 74.62 | 43.00 | 79.76 | 78.47 | 71.43 | – | 65.65 |
| MoNE | $P_{50\%}$ | 16.57GB | 37.20 | 67.17 | 36.80 | 75.30 | 73.39 | 67.88 | – | 59.80 |
| MoNE | $P_{25\%}$ $Q_{4b}$ | 6.00GB | 44.86 | 73.51 | 63.78 | 81.08 | 68.65 | 66.85 | 49.03 | 63.97 |
| He et al. | $P_{50\%}$ | 16.57GB | 33.15 | 58.92 | 57.84 | 78.20 | 50.45 | 62.52 | 36.05 | 53.88 |
| He et al. | $P_{25\%}$ $Q_{4b}$ | 6.00GB | 44.86 | 73.51 | 63.78 | 81.08 | 68.65 | 66.85 | 49.03 | 63.97 |
| C-Prune | $P_{50\%}$ | 16.57GB | 24.40 | 35.65 | 41.14 | 58.92 | 55.87 | 51.22 | 29.52 | 42.39 |
| C-Prune | $P_{25\%}$ $Q_{4b}$ | 6.00GB | 41.72 | 72.81 | 53.53 | 77.86 | 70.98 | 67.80 | 47.97 | 61.81 |
| MoE-I$^2$ | $L_{53.98\%}$ | 15.39GB | 42.58 | 71.8 | 55.16 | – | 76.79 | 67.64 | – | 59.62 |
| **Ours** | $P_{50\%}$ | **16.57GB** | **42.49** | **73.78** | **74.59** | **78.40** | **71.77** | **69.22** | **44.16** | **64.92** |
| **Ours$_Q$** | $P_{25\%}$ $Q_{4b}$ | **6.00GB** | **47.61** | **76.35** | **76.25** | **79.54** | **75.29** | **69.30** | **48.91** | **67.61** |

from 57.24GB to 32.02GB on Qwen3-30B-A3B. Combining $P_{25\%}$ with $Q_{4b}$ achieves the smallest storage, reducing it by over $3\times$, although the throughput may drop slightly due to on-the-fly dequantization. Results for additional MoEs are reported in Appendix Section C.2.3.

### 5.3. Ablation Studies

Table 6 compares inter-layer and intra-layer sparsity allocation under a 50% pruning budget. Simple heuristics (uniform or U-shaped schedules) consistently underperform data-driven strategies, indicating that the expert importance in MoE is highly unbalanced. Coverage-based allocation strategy improves both inter- and intra-layer results. For inter-layer allocation, coverage initialized with smoothed loss performs best (40.0 on ARC-c, 58.2 on GSM8K), approaching the non-pruned model (40.4 and 61.5). For intra-layer allocation, coverage initialized with the attribution-based proxy also outperforms others. These results confirm that coverage-based allocation is robust under aggressive pruning, and attribution-approximated loss yields stronger importance estimates and better performance.

**Smoothing of layerwise loss.** The square-root smoothing is a simple monotone-concave transform for compressing the dynamic range of layerwise losses. As shown in Table 7, all smoothed variants outperform the unsmoothed baseline, and square-root gives the best average. Definitions of all smoothing functions and full per-task results are provided in Appendix Section C.3.3.

**AAR residual reallocation strategy.** In AAR reallocation, we compare two criteria: *largest removed channels* (l-r-c), which prioritizes structural capacity recovery, and *largest removed scores* (l-r-s), which targets score-weighted importance loss. Table 8 suggests that channel count and importance-weighted loss are strongly correlated under coverage-based pruning, both of which yield comparable accuracy across all block sizes $a \in \{64, 128, 256\}$.

### 5.4. Visualization and Analysis

**Coverage ratio vs. Prune ratio.** To examine the coverage-based allocation, we visualize the layer-wise raw loss, score coverage ratio and channel keep ratio in Figure 7. Our method consistently retains a high fraction of

*Table 7.* Comparison of smoothing functions on Qwen1.5-MoE-A2.7B under $P_{50\%}$.

| Smooth fn. | Avg |
|---|---|
| None | 52.40 |
| Log | 56.92 |
| Huber-style | 55.92 |
| Clip | 56.69 |
| **Sqrt (ours)** | **58.27** |

*Table 6.* Comparison of inter- and intra-layer allocation strategies.

| Strategy | ARC-c | GSM8K | HumanEval |
|---|---|---|---|
| *Inter-layer sparsity allocation* | | | |
| Non-prune | 40.4 | 61.5 | 34.2 |
| Uniform | 37.3 | 57.8 | 30.5 |
| U-shaped | 37.2 | 57.6 | 29.3 |
| Smoothed Loss | 38.7 | 55.4 | 28.7 |
| **Coverage-based** | – | – | – |
| ↪ Uniform | 38.0 | 57.0 | 29.3 |
| ↪ Raw loss | 38.4 | 56.1 | 30.5 |
| ↪ **Smoothed Loss** | **40.0** | **58.2** | **30.5** |
| *Intra-layer sparsity allocation* | | | |
| Non-prune | 40.4 | 61.5 | 34.2 |
| Uniform | 38.6 | 52.6 | 22.6 |
| Loss-based | 37.0 | 48.6 | 16.5 |
| Router weight | 38.2 | 57.3 | 22.6 |
| Topk-Channel | 37.7 | 50.6 | 29.9 |
| **Coverage-based** | – | – | – |
| ↪ Uniform | 37.9 | 53.8 | 23.2 |
| ↪ Raw loss | 36.1 | 45.7 | 12.2 |
| ↪ Router logits | 37.5 | 55.2 | 23.8 |
| ↪ **Attribution** | **40.0** | **58.2** | **30.5** |

*Table 8.* Comparison of two AAR residual reallocation strategies on Qwen1.5-MoE-A2.7B with different alignment block sizes $a$. CSQA Avg is the mean accuracy over PIQA, ARC-c, ARC-e, BoolQ, HellaSwag, and WinoGrande.

| Strategy | $a$ | GSM8K | HumanEval | CSQA Avg |
|---|---|---|---|---|
| l-r-c | 64 | 56.29 | 26.83 | 65.12 |
| l-r-c | 128 | 56.29 | 27.44 | 65.44 |
| l-r-c | 256 | 57.29 | 24.39 | 65.47 |
| l-r-s | 64 | 54.69 | 28.05 | 65.14 |
| l-r-s | 128 | 55.89 | 26.83 | 65.04 |
| l-r-s | 256 | 56.69 | 24.39 | 66.40 |

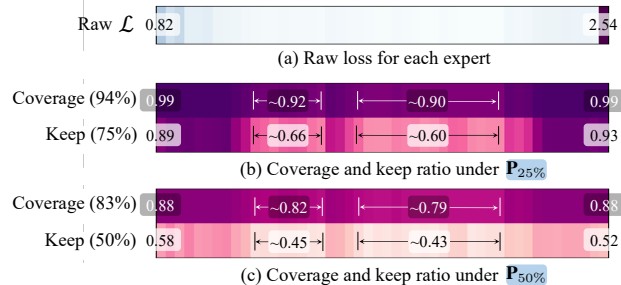

*Figure 7.* Raw loss (top colorbar), score coverage ratio (purple colorbar) vs. channel keep ratio (channels retained after structured pruning, pink colorbar) for each layer on Qwen3-30B-A3B.

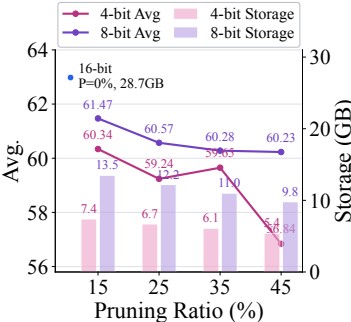

*Figure 8.* Combinations of pruning and quantization. The blue point is the 16-bit baseline at $P = 0\%$.

*Table 9.* Robustness under different top-$k$ routing strategies.

| Top-$k$ | P% | Avg |
|---|---|---|
| **Qwen1.5-MoE-A2.7B** | | |
| 4 | 0 | 62.98 |
| 4 | 50 | 58.27 |
| 2 | 0 | 62.98 |
| 2 | 50 | 58.27 |
| 1 | 0 | 60.56 |
| 1 | 50 | 55.15 |
| **DeepSeek-V2-Lite** | | |
| 6 | 0 | 62.23 |
| 6 | 50 | 59.08 |
| 2 | 0 | 54.84 |
| 2 | 50 | 50.12 |
| 1 | 0 | 62.23 |
| 1 | 50 | 59.08 |

cumulative channel scores by maximizing score coverage (about 90% to 99% under $\mathbf{P}_{25\%}$), while the kept channels vary widely from 60% to 93%. This highlights that layers or experts with highly concentrated scores can preserve most information with relatively few channels. We also provide expert-level visualizations, channel score distributions, and the resulting sparsity allocation in Appendix Section C.5.1.

**Robustness across routing architectures.** Our method is not tied to a specific routing design. The main-text experiments already cover Qwen-style standard top-$k$ routing and the more constrained DeepSeek routing with load balancing. We further switch the activated experts from top-2 to top-1 on Qwen1.5-MoE-A2.7B and DeepSeek-V2-Lite. As shown in Table 9, accuracy drops under top-1 routing, but our method still preserves most of the original performance at 50% pruning under both settings. Full results and router entropy statistics are provided in Appendix Section C.4.2, confirming that expert heterogeneity persists across routing dynamics, layer depths, and data sources.

**Additional analyses.** Appendix Section C.4.1 evaluates calibration-corpus sensitivity and shows that general tasks remain stable across general-domain corpora, while math and code benefit from matched calibration data. We also provide wider pruning–quantization sweeps, second-order attribution comparisons, and AAR hyperparameter studies in Appendix Sections C.2.2, C.3.2 and C.3.4, which support the same accuracy–efficiency and robustness trends.

# 6. Conclusion

We propose an attribution-guided, expert-wise slimming framework for MoEs that reformulates pruning as maximizing channel-score coverage, which better captures internal redundancy and avoids allocating capacity to low-contribution structures. With alignment-aware redistribution, the pruned model remains kernel-compatible for low-bit quantization and achieves substantial compression while preserving accuracy. Experiments on modern MoEs demonstrate a practical path toward efficient MoE deployment.

# Impact Statement

This paper presents work whose goal is to advance the field of machine learning. There are many potential societal consequences of our work, none of which we feel must be specifically highlighted here.

## Acknowledgement

This work was supported by the National Natural Science Foundation of China (Nos. 62476018, 62525601), the Academic Excellence Foundation of BUAA for PhD Students, and the Fundamental Research Funds for the Central Universities.

Dr Tao's research is partially supported by NTU RSR and Start Up Grants.

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

**Contents**

## A. Algorithms

### A.1. Complete Process of Maximum Coverage Allocation Algorithm

This section provides full algorithmic details for the Coverage-Based Allocation (CBA) introduced in Section 4.2, including the concrete inter-layer and intra-layer instantiations and the associated search procedures.

**Inter-layer Allocation.** We first allocate the overall channels to each layer based on layerwise saliency coverage ratio search. Given a target prune ratio $p \in (0, 1)$, we have the overall remaining channels for the prune model $N_{\text{target}} = (1 - p) I E L$, where $I, E, L$ are the intermediate dimension, expert numbers in each layer, and number of layers, respectively.

*Step 1: Channel saliency preparation.* Let $c \in \mathcal{C}_\ell$ represent all the intermediate channels at layer $\ell$ (i.e., $|\mathcal{C}_\ell| = I E$). Each channel has a non-negative saliency score $s_{\ell,c} \geq 0$, which can be computed through various criteria, such as activation magnitude, weight norms, gradient norms, or simple combinations of them. We provide an ablation on the selection of channel saliency criteria in Section C.3.1.

*Step 2: Prefix sums calculation.* We sort channels layerwisely by their saliency scores in descending order, and rewrite $s_{\ell,(1)} \geq s_{\ell,(2)} \geq \cdots \geq s_{\ell,(|\mathcal{C}_\ell|)}$ for the sorted scores. Next, we calculate the prefix sums of the leading $n$ channels at layer $\ell$ as $\mathcal{S}_\ell(n)$ by Algorithm 2:

$$\mathcal{S}_\ell(n) = \sum_{i=0}^{n} s_{\ell,(i)}, \text{ where } 0 \leq n \leq |\mathcal{C}_\ell|. \tag{9}$$

Based on the prefix sums, the saliency coverage ratio of top-$n$ channels in layer $\ell$ can be obtained in $\mathcal{O}(1)$ time as $\rho_\ell(n) = \frac{\mathcal{S}_\ell(n)}{\mathrm{S}_\ell^{tot}}$, where $\mathrm{S}_\ell^{tot} = \sum_{c \in \mathcal{C}_\ell} s_{\ell,c}$ is the total saliency score.

---

**Algorithm 2** Prefix sums $\mathcal{S}_g(n)$ for a group $g$ (a layer or an expert)

1: **Input:** sorted channel scores $s_{g,(1)} \geq s_{g,(2)} \geq \cdots \geq s_{g,(|\mathcal{C}_g|)}$ for group $g$; total channel number $|\mathcal{C}_g|$
2: **Output:** prefix sums $\mathcal{S}_g(n)$ for $n = 1, \ldots, |\mathcal{C}_g|$
3: $\mathcal{S}_g(0) \leftarrow 0$
4: **for** $n = 1$ **to** $|\mathcal{C}_g|$ **do**
5: $\quad \mathcal{S}_g(n) \leftarrow \mathcal{S}_g(n-1) + s_{g,(n)}$
6: **end for**
7: **return** $\mathcal{S}_g(n)$ for all $n = 1, \ldots, |\mathcal{C}_g|$

---

*Step 3: Layerwise loss collection.* Inspired by previous studies that layers have diverse functionalities and redundancy (Skean et al., 2025), we use a small calibration dataset to collect layerwise loss by injecting a scaling as noise into a specific layer, and computing the Negative Log-Likelihood Loss (NLL) compared to the original model. Since the raw loss can have a large range, we conduct a simple smooth by square-root, and get the initial layerwise target saliency coverage ratio $\mathbf{w} \in \mathbb{R}^L$.

*Step 4: Best coverage ratio by binary search.* Given a target global pruning ratio $p$, our goal is to find the largest saliency coverage ratio $\boldsymbol{\rho}^\star \in [0, 1]$ that covers as more high-saliency channels as possible, under the constraint of total remaining channels $N_{\text{target}}$. We conduct an one-dimensional binary search to approach the supremum $\boldsymbol{\rho}^\star$.

As shown in Algorithm 1, we apply a coefficient $\alpha$ as the starting point (#L5):

$$\boldsymbol{\rho}(\alpha) = \min(\alpha \, \boldsymbol{\phi}, 1), \tag{10}$$

where $\boldsymbol{\rho}(\alpha) = \{\rho_1, \rho_2, \ldots, \rho_L\}$. Then, we accumulate the minimal number of channels required in each layer to reach the saliency coverage ratio as

$$N(\boldsymbol{\rho}(\alpha)) = \sum_{\ell \in [1,L]} N_\ell(\rho_\ell) \tag{11}$$

$$= \sum_{\ell \in [1,L]} \min \left\{ n \,\middle|\, \mathcal{S}_\ell(n) \geq \rho_\ell \, \mathrm{S}_\ell^{tot} \right\}. \tag{12}$$

Since $\mathcal{S}_\ell(n)$ is monotonic, we can quickly find $N_\ell(\rho)$ in $O(1)$ time. Algorithm is provided in Algorithm 3.

---

**Algorithm 3** Binary search of minimal channels $N_g(\rho)$ for target coverage $\rho$ in group $g$

1: **Input:** prefix sums $\mathcal{S}_g(n)$; target coverage $\rho \in [0, 1]$; total score $\mathrm{S}_g^{tot}$; channel number $|\mathcal{C}_g|$
2: **Output:** minimal number of channels $N_g(\rho)$
3: $n_{\min} \leftarrow 1$, $n_{\max} \leftarrow |\mathcal{C}_g|$
4: **while** $n_{\min} < n_{\max}$ **do**
5: $\quad n \leftarrow \lfloor (n_{\min} + n_{\max})/2 \rfloor$
6: $\quad$ **if** $\mathcal{S}_g(n) \geq \rho \, \mathrm{S}_g^{tot}$ **then**
7: $\quad\quad n_{\max} \leftarrow n$
8: $\quad$ **else**
9: $\quad\quad n_{\min} \leftarrow n + 1$
10: $\quad$ **end if**
11: **end while**
12: $N_g(\rho) \leftarrow n_{\min}$
13: **return** $N_g(\rho)$

---

*Step 5: Termination condition.* If the gap between the current total number of preserved channels $N(\rho)$ and the target $N_{\text{budget}}$ (#L8) falls below the tolerance $\epsilon$ (to control pruning precision, $\epsilon$ is set to 0.01 as default), the loop is terminated right away (#L9). In practice, we also impose a maximum iterations to prevent long searching. If not terminated, we

perform a binary search over $\alpha \in [0, 1]$ (#L12–16), and finally get the supreme $\alpha^\star$ such that $N(\alpha^\star) \leq N_{\text{budget}} + \epsilon N^{tot}$, i.e., $p(\boldsymbol{\rho}^\star) \approx p$.

**Intra-Layer Allocation.** After layerwise budget $N_\ell^\star$ is fixed, we perform the coverage search for intra-layer pruning for each expert.

---

**Algorithm 4** Intra-layer coverage search at layer $\ell$ with an expert-wise importance prior

1: **Input:** experts $\mathcal{E}_\ell = \{1, \dots, E\}$ at layer $\ell$; non-negative importance prior $\{\phi_{\ell,e}\}_{e\in\mathcal{E}_\ell}$; prefix sums $\{\mathcal{S}_{\ell,e}(n), \mathrm{S}_{\ell,e}^{tot}\}_{e\in\mathcal{E}_\ell}$;
   per-expert channel counts $\{|\mathcal{C}_{\ell,e}|\}_{e\in\mathcal{E}_\ell}$; layer budget $N_\ell^\star$; tolerance $\varepsilon$
2: **Output:** expert-wise channel budgets $\{N_{\ell,e}^\star\}_{e\in\mathcal{E}_\ell}$
3: $N_\ell^{tot} \leftarrow \sum_{e\in\mathcal{E}_\ell} |\mathcal{C}_{\ell,e}|$
4: $\alpha_{\min} \leftarrow 0, \; \alpha_{\max} \leftarrow 1$
5: **while** $\alpha_{\min} < \alpha_{\max}$ **do**
6:     $\alpha \leftarrow (\alpha_{\min} + \alpha_{\max})/2$
7:     **for** each expert $e \in \mathcal{E}_\ell$ **do**
8:         $\rho_{\ell,e} \leftarrow \min(\alpha\phi_{\ell,e}, 1)$
9:         $N_{\ell,e}(\rho_{\ell,e}) \leftarrow \min\{n \mid \mathcal{S}_{\ell,e}(n) \geq \rho_{\ell,e}\, \mathrm{S}_{\ell,e}^{tot}\}$                      (Algorithm 3)
10:    **end for**
11:    $N_\ell(\boldsymbol{\rho}) \leftarrow \sum_{e\in\mathcal{E}_\ell} N_{\ell,e}(\rho_{\ell,e})$
12:    **if** $\left|N_\ell(\boldsymbol{\rho}) - N_\ell^\star\right| \leq \varepsilon N_\ell^{tot}$ **then**
13:       $N_{\ell,e}^\star \leftarrow N_{\ell,e}(\rho_{\ell,e}), \forall e \in \mathcal{E}_\ell$
14:       **break**
15:    **end if**
16:    **if** $N_\ell(\boldsymbol{\rho}) > N_\ell^\star$ **then**
17:       $\alpha_{\max} \leftarrow \alpha$
18:    **else**
19:       $\alpha_{\min} \leftarrow \alpha$
20:    **end if**
21: **end while**
22: **return** $\{N_{\ell,e}^\star\}_{e\in\mathcal{E}_\ell}$

---

*Step 1: Channel saliency reuse.* We reuse the channel saliency scores obtained above and denote as $s_{\ell,e,c}$ for expert $e$ at layer $\ell$. Next, since we have calculated the layerwise prefix sums, it only costs $<0.1ms$ to recompute the expert-wise prefix sums $\mathcal{S}_{\ell,e}(n)$. Time breakdown can be found in Appendix Section C.2.4.

*Step 2: Expert-wise importance estimator.* We provide an efficient and accurate expert-wise loss approximation in Section 4.1, taking place of the time-consuming expert-wise loss collection by ablating a specific expert one-by-one. Let $\phi_\ell$ denote the expert-wise importance at layer $\ell$.

*Step 3: Best coverage ratio by binary search.* Similar to inter-layer saliency coverage search, given a target number of remaining channels $N_\ell^\star$, we start from an initial scaling factor $\alpha \in (0, 1)$ and have

$$\boldsymbol{\rho}_\ell(\alpha) = \min\left(\alpha\,\boldsymbol{\phi}_\ell, 1\right), \qquad \forall \ell \in [1, L], \tag{13}$$

where $\boldsymbol{\rho}_\ell(\alpha) = \{\rho_{\ell,1}, \rho_{\ell,2}, \dots, \rho_{\ell,E}\}$. Next, the minimal channels required at layer $\ell$ to reach the coverage is

$$N_\ell(\boldsymbol{\rho}_\ell(\alpha)) = \sum_{e\in[1,E]} N_{\ell,e}(\rho_{\ell,e}) \tag{14}$$

$$= \sum_{e\in[1,E]} \min\left\{n \mid \mathcal{S}_{\ell,e}(n) \geq \rho_{\ell,e}\, S_{\ell,e}\right\}, \tag{15}$$

where $\mathrm{S}_{\ell,e}^{tot} = \sum_{c\in\mathcal{C}_{\ell,e}} s_{\ell,e,c}$ is the total saliency score of expert $e$ at layer $\ell$. We then iteratively search $\alpha$ to adjust $\boldsymbol{\rho}_\ell$, comparing $N_\ell(\boldsymbol{\rho}_\ell)$ against the target channel budget $N_\ell^\star$, until we find the optimal $\alpha^\star$ that $\left|N_\ell(\boldsymbol{\rho}_\ell^\star) - N_\ell^\star\right| \leq \varepsilon N_\ell^{tot}$, where $\boldsymbol{\rho}_\ell^\star = \alpha^\star \mathbf{w}_\ell, \forall \ell \in [1, L]$. Each layer repeats the same binary search. The complete algorithm for intra-layer allocation can be found in Algorithm 4.

Combining inter-layer and intra-layer allocation, we obtain the final pruning budget for each expert $N_{\ell,e}^{\star}$ that satisfy the global pruning constraint $p$ while maximizing the saliency coverage ratio with the given layerwise/expert-wise importance. In this way, layers/experts that have more redundancy will have a smaller budget, on the contrary, layers/experts that have dispersed channel saliency can have more remaining channels.

As for the computational complexity of the overall allocation process, since $\mathcal{S}_{\ell}(n)$ and $\mathcal{S}_{\ell,e}(n)$ are non-decreasing in $n$, $N(\boldsymbol{\rho}(\alpha))$ and $N_{\ell}(\boldsymbol{\rho}_{\ell}(\alpha))$ are non-decreasing in $\alpha$, the binary search can be efficiently performed over $\alpha$ in $\mathcal{O}(1)$ time to find the optimal $\alpha^{\star}$ for global and for all the layers.

### A.2. Hamilton apportionment redistribution.

We regard the channel redistribution problem as the classical *Hamilton apportionment*. Let $N_{\ell,e}$ denote the allocated channels of expert $e$ in layer $l$ given by our maximized coverage algorithm introduced above, and let $a$ be the supported GEMM block size (e.g., $a = 64, 128, \ldots$). We additionally introduce a minimal channel threshold $m$ to eliminate extremely small experts that carries little information due to too few channels left.

*Step 1: Minimal-channel trimming.* We first trim experts whose allocated channels are smaller than $m$:

$$\tilde{N}_{\ell,e} = \begin{cases} 0, & N_{\ell,e} < m, \\ N_{\ell,e}, & N_{\ell,e} \geq m, \end{cases} \qquad \mathcal{A}_{\ell} = \{e \mid \tilde{N}_{\ell,e} > 0\}, \tag{16}$$

where $\mathcal{A}_{\ell}$ denotes the active experts after trimming.

*Step 2: Downward alignment.* For each remaining expert $e \in \mathcal{A}_{\ell}$, we round $\tilde{N}_{l,e}$ down to the nearest multiple of $a$, ensuring compatibility with low-bit GEMM kernels:

$$N_{\ell,e}^{\text{base}} = \left\lfloor \frac{\tilde{N}_{\ell,e}}{a} \right\rfloor \cdot a, \qquad e \in \mathcal{A}_{\ell}, \tag{17}$$

and set $N_{\ell,e}^{\text{base}} = 0$ for trimmed experts $e \notin \mathcal{A}_{\ell}$.

*Step 3: Compute remaining quota and available blocks.* The channel budget released by trimming and alignment is collected and segmented as units of $a$-blocks. Therefore, the remaining quota to be redistributed , and the number of available blocks in layer $\ell$ can be expressed as

$$R_{\ell} = N_{\ell}^{\star} - \sum_{e \in [1,E]} N_{\ell,e}^{\text{base}}, \quad q_{\ell} = \left\lfloor \frac{R_{\ell}}{a} \right\rfloor. \tag{18}$$

*Step 4: Hamilton apportionment over experts.* We redistribute the remaining quota in $q_{\ell}$ discrete $a$-blocks.

The fractional remainder of each expert induced by downward alignment is:

$$r_{\ell,e} = \frac{\tilde{N}_{\ell,e} - N_{\ell,e}^{\text{base}}}{a} \in [0,1), \quad e \in \mathcal{A}_{\ell}. \tag{19}$$

To approach the original allocation derived by the expert importance, each expert can receive at most one additional block. We sort $r_{\ell,e}$ in descending order, and let $\pi$ be a permutation of $\mathcal{A}_{\ell}$ such that $r_{\ell,\pi(1)} \geq r_{\ell,\pi(2)} \geq \cdots \geq r_{\ell,\pi(|\mathcal{A}_{\ell}|)}$. The largest $q_{\ell}$ experts can receive the additional $a$-block, which can be simply written as

$$b_{\ell,e} = \mathbb{I}\left[e \in \{\pi(1), \ldots, \pi(q_{\ell})\}\right], \tag{20}$$

Finally, the aligned channels are

$$N_{\ell,e}' = N_{\ell,e}^{\text{base}} + a \cdot b_{\ell,e}. \tag{21}$$

The resulting aligned channels approaches the original layerwise allocation budget, satisfies expert capacity constraints, and guarantees that every expert has a channel dimension divisible by $a$. It enables the pruned model to be stored and computed by low-bit quantization, yielding both effective compression and inference speedup without redundant zero padding on MoE models.

## B. Derivation and Proof

### B.1. Complete Proof of Attribution-based Loss Approximation in Section 4.1

Let $h_\ell \in \mathbb{R}^d$ be the input hidden state of the MoE block in layer $\ell$, and let $z_{\ell,e} = f_{\ell,e}(h_\ell) \in \mathbb{R}^d$ denote the output of expert $e$ before gating. The output of MoE layer $\ell$ is the weighted sum of top-$k$ experts:

$$y_\ell = \sum_{e \in \mathcal{E}_\ell} g_{\ell,e}(h_\ell)\, z_{\ell,e}. \tag{22}$$

where $\mathcal{E}_\ell$ is the top-$k$ experts selected by the router at layer $\ell$, and $|\mathcal{E}_\ell| = k$ ($k$ is typically set as $1, 2, 4, 8$ in modern MoE). $g_{\ell,e}(h_\ell) \geq 0$ is the router weight of expert $e$.

We measure the contribution of expert $e$ at layer $\ell$ by the loss change when removing this expert. If the expert $e \in \mathcal{E}_\ell$ is ranked as top-$k$ by the router and selected for a specific token, removing it corresponds to replacing $z_{\ell,e}$ with zero, which will induce a perturbation in the layer output

$$\Delta y_\ell^{(e)} = \hat{y}_\ell^{(e)} - y_\ell \tag{23}$$

$$= \sum_{e' \in \mathcal{E}_\ell \setminus \{e\}} g_{\ell,e'} z_{\ell,e'} - \sum_{e' \in \mathcal{E}_\ell} g_{\ell,e'} z_{\ell,e'} \tag{24}$$

$$= -\, g_{\ell,e}\, z_{\ell,e}, \tag{25}$$

where $\hat{y}_\ell^{(e)}$ is the layer output when removing expert $e$.

Let $\mathcal{L}$ be the loss compared to the original layer's output. For any perturbation $\Delta y$ applied to the layer output $y_\ell$, the loss can be written by the first-order Taylor expansion as

$$\mathcal{L}(y_\ell + \Delta y) = \mathcal{L}(y_\ell) + \left(\frac{\partial \mathcal{L}}{\partial y_\ell}\right)^\top \Delta y + \mathcal{O}(\|\Delta y\|^2). \tag{26}$$

In our case, removing expert $e$ at layer $\ell$ induces $\Delta y_\ell^{(e)}$ (Equation (23)) to the block output, and the loss change is

$$\Delta \mathcal{L}^{(e)} = \mathcal{L}(y_\ell + \Delta y_\ell^{(e)}) - \mathcal{L}(y_\ell), \tag{27}$$

which can be approximated by only keeping the first-order term in Equation (26) as

$$\Delta \mathcal{L}^{(e)} \approx \left(\frac{\partial \mathcal{L}}{\partial y_\ell}\right)^\top \Delta y_\ell^{(e)} = -\left(\frac{\partial \mathcal{L}}{\partial y_\ell}\right)^\top (g_{\ell,e} z_{\ell,e}). \tag{28}$$

By the chain rule, the gradient w.r.t. the expert output $z_{\ell,e}$ is

$$\frac{\partial \mathcal{L}}{\partial z_{\ell,e}} = g_{\ell,e}\, \frac{\partial \mathcal{L}}{\partial y_\ell}, \tag{29}$$

so the final loss change can be estimated as

$$\Delta \mathcal{L}^{(e)} \approx -\left(\frac{\partial \mathcal{L}}{\partial z_{\ell,e}}\right)^\top z_{\ell,e}. \tag{30}$$

The approximated loss is then used to measure the importance of experts at layer $\ell$ altogether.

### B.2. Derivation of expected redundant channels in Section 4.3 Rationale

From the pruning perspective, one can define a purely logical sparsity level as

$$s_{\text{logical}} = 1 - \frac{K}{D},$$

where $D$ is the original channel dimensionality and $K$ is the number of channels retained after pruning. Under 4-bit quantization, however, parameters are physically stored and processed in fixed-size blocks. With a block size of 64, a linear layer with effective hidden size $K$ is packed as

$$\tilde{D} = \left\lceil \frac{K}{64} \right\rceil \cdot 64,$$

and the corresponding physical compression ratio becomes

$$s_{\text{physical}} = 1 - \frac{\tilde{D}}{D}.$$

If we do not explicitly align $K$ during pruning, each expert can waste between 0 and 63 channels at the storage level. Assuming that the residue $K \bmod 64$ is approximately uniform in $\{0, \ldots, 63\}$, the expected padding overhead per expert is

$$\mathbb{E}[\tilde{D} - K] = \frac{1}{64} \sum_{r=1}^{63} (64 - r) = 31.5 \text{ channels.}$$

For a Qwen3-style MoE block with hidden size $D = 768$, $E = 128$ experts and $L = 64$ layers, this corresponds to roughly

$$\frac{31.5}{768} \approx 4.1\%$$

## C. Experiments

### C.1. Experimental Setup

**Models.**  We conduct experiments on the following representative open-source MoE LLMs that cover different scales and architectural choices: DeepSeek-MoE-16B (Dai et al., 2024), DeepSeek-V2-Lite (DeepSeek-AI, 2024), Qwen1.5-MoE-A2.7B (Team, 2024), and Qwen3-30B-A3B-Thinking (Qwen-Team, 2025).

**Compared Methods.**  We compare our approach with advanced MoE compression baselines with various techniques: EAC-MoE (Chen et al., 2025a) performs joint pruning and quantization; we report its configurations with $\alpha = 0.3$ (11% sparsity) and $\alpha = 0.7$ (38% sparsity), and the corresponding average bitwidth of 3.03. He et al. (He et al., 2025) combines expert trimming and slimming; we report the 25% layer or block drop setting together with 4-bit AWQ quantization. MoE-I$^2$ (Yang et al., 2024) jointly applies inter-expert pruning to remove redundant experts and intra-expert low-rank decomposition to reduce the parameter redundancy within remaining expert. PuzzleMoE (Zhao et al., 2025) focuses on expert merging by 25% or 50%, and provides customized CUDA kernels for efficient inference. MoNE (Zhang et al., 2025) prunes MoE models by replacing redundant experts with lightweight counterparts. C-Prune (Guo et al., 2025b) addresses intra-layer and inter-layer expert redundancy in MoE LLMs via a two-stage framework of layer-wise expert clustering followed by global cluster pruning. Wanda (Sun et al., 2023) is a training-free unstructured pruning method that scores each weight by the product of its magnitude and the corresponding input activation norm, requiring no retraining or weight reconstruction. We report the results from the original papers under the closest comparable settings.

**Pruning Settings.**  We use channel pruning as structural sparsification technique for easy implementation by mainstream inference engine. In the following experiments, we adopt two variants: **Ours** applies 50% channel sparsity without quantization, and thus does not require alignment-aware redistribution. If an expert is assigned zero channel after pruning, we trim the expert and shrink the corresponding router dimension, so that the expert is never selected. Furthermore, **Ours**$_Q$ applies 25% channel sparsity and further performs 4-bit quantization using BitsAndBytes NF4. Meanwhile, we enable Alignment-Aware Redistribution when applied quantization with granularity $a = 128$ and enforce the minimum expert channel size $m = 128$. We select $a = 128$ and $m = 128$ based on a small grid search over feasible settings, constrained by linear layer shapes and quantized operator support. We report throughput and peak memory trade-offs of the explored settings in Appendix Section C.2.3, Figure 10, and choose the setting with best overall efficiency.

**Calibration and Fine-tuning.**  We use C4 (Raffel et al., 2019) as the calibration dataset for commonsense tasks. For reasoning benchmarks, we calibrate using samples drawn from GSM8K (Cobbe et al., 2021) or OpenCodeReasoning (Ahmad et al., 2025) depending on the task category. After pruning, we follow Yang et al. to perform fine-tuning on Alpaca (Taori

et al., 2023) for 2 epochs. We fine-tune the MoE blocks using DoRA (Liu et al., 2024b) with rank 32 and learning rate $1e-4$, while adapting the routing module with rank 4 and learning rate $1e-6$. We use AdamW with warmup ratio 0.1 and clip gradient exceeding 0.5, without weight decay. All training is conducted on $4\times$H20 GPUs. The training cost is 12 GPU hours for Qwen1.5-MoE-A2.7B and 48 GPU hours for Qwen3-30B-A3B, and models of similar scale exhibit comparable training time.

**Benchmarks and Evaluation.** We evaluate using two widely adopted toolkits: the LM Evaluation Harness[2] and OpenCompass[3]. We report zero-shot performance on general reasoning and knowledge benchmarks, including ARC-C (Clark et al., 2018), ARC-E (Clark et al., 2018), HellaSwag (Zellers et al., 2019), PIQA (Bisk et al., 2019), BoolQ (Clark et al., 2019), WinoGrande (Sakaguchi et al., undefined), and MMLU (Hendrycks et al., 2020), and math/code benchmarks with 8-shot, including GSM8K (Cobbe et al., 2021), HumanEval (Chen et al., 2021), MATH500 (Lightman et al., 2023), AIME25 (**?**), GPQA (**?**), and LiveCodeBench (Jain et al., 2024). We follow the default task configurations and official evaluation protocols provided by the toolkits and report standard metrics, e.g., *accuracy* for multiple-choice tasks, *exact match* for math, and *pass@1* for code generation. For long-context evaluations, we set the maximum sequence and output lengths as follows: AIME25 uses `MAX_SEQ_LEN=65536` and `MAX_OUT_LEN=32768`; MATH500 uses `MAX_SEQ_LEN=16384` and `MAX_OUT_LEN=4096`; LiveCodeBench_v6_academic uses `MAX_SEQ_LEN=32768` and `MAX_OUT_LEN=16384`.

## C.2. Overall Comparisons

### C.2.1. PARETO FRONTIER OF CHANNEL-LEVEL VS. EXPERT-LEVEL PRUNING METHODS

We provide the full per-task accuracy for channel-level pruning and expert-level pruning baselines under matched storage budgets. Across moderate-to-aggressive budgets (25%–75%), channel-level pruning consistently stays on the Pareto frontier in Figure 9. Under the mildest 13.3% pruning setting, the channel budget is loose enough that expert-level pruning remains competitive, but the advantage of channel-level allocation becomes pronounced once the compression budget tightens.

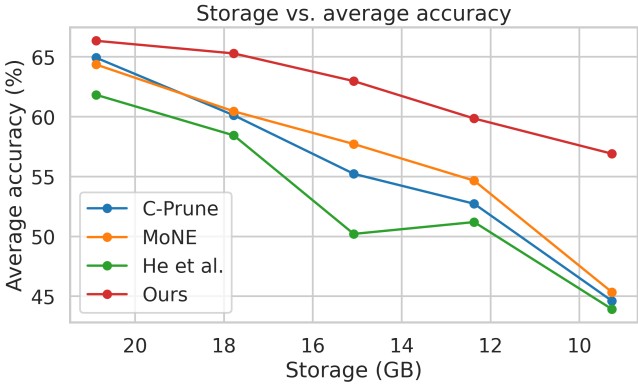

*Figure 9.* Pareto frontier of average downstream-task accuracy versus compressed model storage (GB) for Qwen1.5-MoE-A2.7B. Our channel-level pruning consistently dominates expert-level baselines across the full compression range.

### C.2.2. WIDER PRUNING–QUANTIZATION COMBINATIONS

Based on our current experiments, $\mathbf{P}_{25\%}$ $\mathbf{Q}_{4b}$ gives the best accuracy-efficiency tradeoff among the default deployment-oriented settings, but we do not claim it is a universal global optimum. We sweep a wider range of $P/Q$ combinations on Qwen1.5-MoE-A2.7B in Table 11. Stronger compression gives lower storage but larger accuracy drop, while milder compression preserves accuracy better. The framework therefore supports flexible operating points depending on deployment constraints.

---

[2]https://github.com/EleutherAI/lm-evaluation-harness
[3]https://github.com/open-compass/opencompass

*Table 10.* Per-task accuracy of channel-level (Ours) vs. expert-level pruning baselines on Qwen1.5-MoE-A2.7B at matched storage budgets. Our method performs channel-level structural pruning, whereas the competing baselines adopt expert-level pruning.

| Method | Expert-level | Prune ratio (%) | Storage (GB) | ARC-c | ARC-e | HellaSwag | PIQA | BoolQ | WinoGrande | Avg |
|---|---|---|---|---|---|---|---|---|---|---|
| C-Prune | ✓ | 13.3 | 23.59 | 44.71 | 69.15 | 77.26 | 80.36 | 79.48 | 68.90 | **69.98** |
| MoNE | ✓ | 13.3 | 23.59 | 41.55 | 63.68 | 77.07 | 80.03 | 76.51 | 68.51 | 67.89 |
| He et al. | ✓ | 13.3 | 23.59 | 40.54 | 65.95 | 63.78 | 82.70 | 74.05 | 67.57 | 65.77 |
| Ours | ✗ | 13.3 | 23.59 | 42.70 | 65.95 | 65.59 | 80.90 | 76.40 | 70.81 | 67.06 |
| C-Prune | ✓ | 25.0 | 20.88 | 40.00 | 62.70 | 63.06 | 78.92 | 77.12 | 67.75 | 64.93 |
| MoNE | ✓ | 25.0 | 20.88 | 40.44 | 60.73 | 64.14 | 81.20 | 71.53 | 68.11 | 64.36 |
| He et al. | ✓ | 25.0 | 20.88 | 37.30 | 59.64 | 61.80 | 81.08 | 67.93 | 63.24 | 61.83 |
| **Ours** | ✗ | **25.0** | **20.88** | **42.16** | **65.41** | **63.24** | **79.10** | **78.20** | **69.91** | **66.34** |
| C-Prune | ✓ | 38.3 | 17.78 | 34.59 | 58.02 | 60.72 | 76.22 | 69.55 | 61.62 | 60.12 |
| MoNE | ✓ | 38.3 | 17.78 | 35.50 | 52.61 | 64.32 | 80.36 | 64.50 | 65.41 | 60.45 |
| He et al. | ✓ | 38.3 | 17.78 | 34.41 | 55.86 | 58.74 | 77.12 | 64.86 | 59.64 | 58.44 |
| **Ours** | ✗ | **38.3** | **17.78** | **40.72** | **64.14** | **63.24** | **78.92** | **75.68** | **69.01** | **65.29** |
| C-Prune | ✓ | 50.0 | 15.08 | 30.63 | 50.09 | 57.12 | 74.23 | 62.16 | 57.12 | 55.23 |
| MoNE | ✓ | 50.0 | 15.08 | 32.07 | 51.71 | 60.54 | 76.58 | 63.60 | 61.80 | 57.72 |
| He et al. | ✓ | 50.0 | 15.08 | 25.95 | 45.41 | 44.86 | 69.37 | 63.06 | 52.61 | 50.21 |
| **Ours** | ✗ | **50.0** | **15.08** | **36.76** | **65.95** | **61.98** | **75.32** | **69.37** | **68.47** | **62.98** |
| C-Prune | ✓ | 61.7 | 12.37 | 28.29 | 44.50 | 53.87 | 69.73 | 65.95 | 54.05 | 52.73 |
| MoNE | ✓ | 61.7 | 12.37 | 30.27 | 46.13 | 55.68 | 73.15 | 63.42 | 59.28 | 54.66 |
| He et al. | ✓ | 61.7 | 12.37 | 27.57 | 45.05 | 47.21 | 72.07 | 61.26 | 54.05 | 51.20 |
| **Ours** | ✗ | **61.7** | **12.37** | **33.15** | **58.20** | **60.00** | **73.87** | **67.39** | **66.49** | **59.85** |
| C-Prune | ✓ | 75.0 | 9.27 | 23.42 | 35.32 | 42.52 | 59.46 | 54.41 | 52.61 | 44.62 |
| MoNE | ✓ | 75.0 | 9.27 | 25.95 | 36.58 | 41.62 | 61.44 | 52.97 | 53.51 | 45.35 |
| He et al. | ✓ | 75.0 | 9.27 | 21.80 | 36.04 | 37.48 | 60.36 | 54.23 | 53.69 | 43.93 |
| **Ours** | ✗ | **75.0** | **9.27** | **32.61** | **52.07** | **56.22** | **69.37** | **67.57** | **63.60** | **56.91** |

### C.2.3. SPEEDUP AND MEMORY USAGE WITH DIFFERENT ALIGNMENT GRANULARITY

Figure 10 reports throughput (tokens/s) and runtime peak memory of Qwen1.5-MoE-A2.7B as a function of the minimum kept-channel threshold $m$ (rows) and the alignment block size $a$ (columns) used in our Alignment-Aware Redistribution.

Across all settings, combining channel pruning with 4-bit quantization substantially reduces peak memory compared to the unpruned baseline, confirming that AAR successfully integrates structural sparsity with low-bit storage. Larger block sizes $a$ align channel counts to coarser multiples, which enables larger and more regular GEMM kernels and is accordingly reflected in higher throughput. However, coarser alignment reduces the degree to which each expert's channel count tracks the original CBA solution, so there is a natural trade-off between kernel efficiency and allocation fidelity.

The minimum-channel threshold $m$ controls the smallest permissible expert width after alignment. Small $m$ allows very thin experts to survive, which can hurt throughput due to irregular kernel sizes, while large $m$ forces low-importance experts to retain more channels than necessary, slightly increasing memory. The heatmap shows that $m = 128$ with $a = 128$ or $a = 256$ achieves a favorable balance: throughput is near the maximum achievable value, and memory remains well below the unpruned baseline. These are the default settings used throughout the main experiments.

### C.2.4. CALIBRATION RUNTIME BREAKDOWN

We provide a time breakdown of the calibration process in Table 12, demonstrating the efficiency of the proposed method. Even for a large MoE model such as Qwen3-30B-A3B, the total calibration time remains within 10 seconds.

## C.3. Further Ablation Studies on Proposed Methods

### C.3.1. CHANNEL SCORE METRIC SELECTION

In this ablation, we only change the channel score definition ($s_{\ell,c}$) while keeping all other components of the pipeline fixed.

**Definition of metrics.**

*Table 11.* Storage and downstream accuracy under different combinations of pruning ratio $P$ and quantization bitwidth $Q$ on Qwen1.5-MoE-A2.7B. The default Ours$_Q$ configuration is highlighted.

| P (%) | Q (bits) | Storage (GB) | PIQA | ARC-c | ARC-e | BoolQ | HellaSwag | WinoGrande | GSM8K | HumanEval | Avg |
|---|---|---|---|---|---|---|---|---|---|---|---|
| 0 | 16 | 28.70 | 80.79 | 40.41 | 69.44 | 70.57 | 77.17 | 69.77 | 61.50 | 34.20 | 62.98 |
| 15 | 8 | 13.50 | 80.04 | 41.52 | 65.07 | 77.64 | 65.07 | 70.26 | 61.08 | 31.10 | 61.47 |
| 25 | 8 | 12.25 | 79.04 | 41.52 | 64.67 | 76.85 | 63.67 | 69.66 | 62.35 | 26.83 | 60.57 |
| 35 | 8 | 11.01 | 78.24 | 39.32 | 61.68 | 75.25 | 63.87 | 67.47 | 62.28 | 34.15 | 60.28 |
| 45 | 8 | 9.76 | 76.65 | 39.32 | 64.07 | 72.85 | 62.28 | 69.46 | 62.48 | 34.76 | 60.23 |
| 15 | 4 | 7.36 | 79.24 | 38.72 | 63.87 | 76.45 | 64.27 | 70.86 | 61.26 | 28.05 | 60.34 |
| **25** | **4** | **6.71** | **79.04** | **40.92** | **60.88** | **76.85** | **63.27** | **69.26** | **56.24** | **27.44** | **59.24** |
| 35 | 4 | 6.07 | 78.24 | 37.92 | 62.28 | 77.05 | 62.87 | 70.26 | 56.89 | 31.71 | 59.65 |
| 45 | 4 | 5.43 | 74.65 | 35.13 | 62.48 | 71.86 | 61.28 | 70.46 | 52.61 | 26.22 | 56.84 |

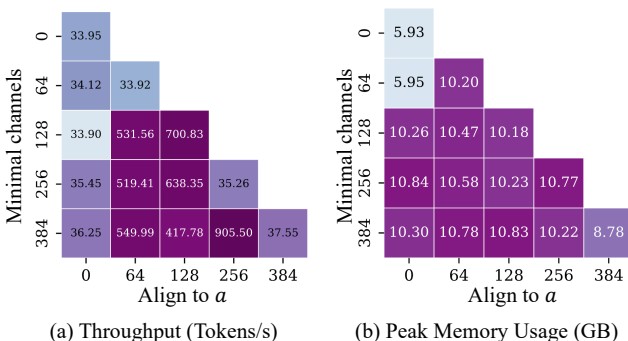

(a) Throughput (Tokens/s)    (b) Peak Memory Usage (GB)

*Figure 10.* Throughput and runtime memory usage of Qwen1.5-MoE-A2.7B with different minimal channel numbers and alignment granularity.

- **Weight Magnitude (channel-wise L2 norm).** For expert $e$ in layer $\ell$ and a projection $\phi$ with weight matrix $W_{\ell,e}^{(\phi)} \in \mathbb{R}^{O_\phi \times I_\phi}$, we define the importance of input channel $c$ by the L2 norm of the corresponding weight column:

$$s_{\ell,e,c}^{(\phi,W)} = \|W_{\ell,e,c}^{(\phi)}\|_2^{O_\phi} = \Big( \sum_{o \in O_\phi} \big( W_{\ell,e,o,c}^{(\phi)} \big)^2 \Big)^{\frac{1}{2}}. \tag{31}$$

- **Activation Magnitude (channel-wise L2 norm).** For expert $e$ in layer $\ell$ and a projection $\phi$, let $A_{\ell,e,t,c}^{(\phi)}$ denote the activation of channel $c$ at token $t$. We compute the channel magnitude by an L2 norm over $\mathcal{T}$ tokens:

$$s_{\ell,e,c}^{(\phi,A)} = \|A_{\ell,e,c}^{(\phi)}\|_2^{\mathcal{T}} = \Big( \sum_{t \in \mathcal{T}} \big( A_{\ell,e,t,c}^{(\phi)} \big)^2 \Big)^{\frac{1}{2}}. \tag{32}$$

- **Weight×Activation (Sun et al., 2023).** We follow Wanda to combine the magnitude of weight with per-channel activation. We compute the inner-product, and then reducing along the output dimension.

$$s_{\ell,e,c}^{(\phi, WA)} = \sum_{o \in O_\phi} \big( |W_{\ell,e,o,c}^{(\phi)}| \cdot \|A_{\ell,e,c}^{(\phi)}\|_2^{\mathcal{T}} \big). \tag{33}$$

- **Gradient Saliency Map (Song et al., 2019).** We use a small calibration set, and collect the activation gradient by forward and backward propagation. Let $g_{\ell,e,t,c}^{(\phi,A)} = \nabla_{A_{\ell,e,t,c}^{(\phi)}} \mathcal{L}$ denote the gradient w.r.t. the activation of channel $c$ in expert $e$ and projection $\phi$ at token $t$. We aggregate the gradients with an L2 norm over the token dimension:

$$s_{\ell,e,c}^{(\phi,g)} = \|g_{\ell,e,c}^{(\phi,A)}\|_2^{\mathcal{T}} = \Big( \sum_{t \in \mathcal{T}} \big( g_{\ell,e,t,c}^{(\phi,A)} \big)^2 \Big)^{\frac{1}{2}}. \tag{34}$$

*Table 12.* Time breakdown of generating prune allocation of 50% sparsity.

| Stage | Time (ms) | Time (%) |
|---|---|---|
| **Qwen1.5-MoE-A2.7B** | | |
| Overall generating prune plan | 2078.31 | 100% |
| – *Inter-layer coverage search* | 63.00 | 3.03% |
| Smooth weights | 30.35 | 1.46% |
| Compute prefix sum | 14.16 | 0.68% |
| Binary search | 17.90 | 0.86% |
| – *Intra-layer coverage search* | 1836.49 | 88.36% |
| Recompute prefix sum | 0.08 | <0.01% |
| Binary Search (24 MoE layers) | 1836.41 | 88.36% |
| – per layer | 76.52 | 3.68% |
| – *Alignment-aware redistribution* | 178.82 | 8.60% |
| Compute $K^{\text{base}}$ | 52.84 | 2.54% |
| Compute headroom | 14.96 | 0.72% |
| Allocate chunks | 110.14 | 5.30% |
| Clamp to $I$ | 0.88 | 0.04% |
| **Deepseek-MoE-16B** | | |
| Overall generating prune plan | 2914.31 | 100% |
| – *Inter-layer coverage search* | 73.81 | 2.53% |
| Smooth weights | 30.48 | 1.05% |
| Compute prefix sum | 15.57 | 0.53% |
| Binary search | 27.14 | 0.93% |
| – *Intra-layer coverage search* | 2647.70 | 90.85% |
| Recompute prefix sum | 0.10 | <0.01% |
| Binary Search (27 MoE layers) | 2647.60 | 90.85% |
| – per layer | 98.06 | 3.36% |
| – *Alignment-aware redistribution* | 192.80 | 6.62% |
| Compute $K^{\text{base}}$ | 56.12 | 1.93% |
| Compute headroom | 17.16 | 0.59% |
| Allocate chunks | 118.57 | 4.07% |
| Clamp to $I$ | 0.95 | 0.03% |
| **Qwen3-30B-A3B** | | |
| Overall generating prune plan (total) | 10063.12 | 100% |
| – *Inter-layer coverage search* | 87.43 | 0.87% |
| Smooth weights | 29.36 | 0.29% |
| Compute prefix sum | 19.91 | 0.20% |
| Binary search | 37.15 | 0.37% |
| – *Intra-layer coverage search* | 9619.28 | 95.59% |
| Recompute prefix sum | 0.09 | <0.01% |
| Binary Search (MoE 48 layers) | 9619.19 | 95.59% |
| – per layer | 200.40 | 1.99% |
| – *Alignment-aware redistribution* | 356.41 | 3.54% |
| Compute $K^{\text{base}}$ | 20.44 | 0.20% |
| Compute headroom | 17.02 | 0.17% |
| Allocate chunks | 317.40 | 3.15% |
| Clamp to $I$ | 1.55 | 0.02% |

- **Activation×Gradient Saliency Map (Song et al., 2019).** We use the element-wise product between activation and its gradient to score channel importance. We compute the absolute value and then average over token dimension:

$$s_{\ell,e,c}^{(\phi, gA)} = \frac{1}{|\mathcal{T}|} \sum_{t \in \mathcal{T}} \left| A_{\ell,e,t,c}^{(\phi)} \cdot g_{\ell,e,t,c}^{(\phi,A)} \right|. \tag{35}$$

- **SNIP First-order Sensitivity (Weight×Gradient) (Lee et al., 2018).** We follow SNIP to score channels by the first-order Taylor approximation, using the element-wise product between weight and its gradient. Let $g_{\ell,e,c}^{(\phi,W)} = \nabla_{W_{\ell,e,o,c}^{(\phi)}} \mathcal{L}$ denote the gradient w.r.t. the weight of input channel $c$ in expert $e$ and projection $\phi$ and output channel $o$, which is obtained by backpropagating the loss on the calibration set. We compute the absolute value and then reduce along the output dimension:

$$s_{\ell,e,c}^{(\phi, Wg)} = \sum_{o \in O_\phi} \left| W_{\ell,e,o,c}^{(\phi)} \cdot g_{\ell,e,o,c}^{(\phi,W)} \right|. \tag{36}$$

*Table 13.* Ablation on channel score definition $s_{\ell,c}$. Inter-layer and intra-layer allocation use the same maximum-coverage procedure, while only the channel saliency scores differ.

| $s_{\ell,c}$ | ARC-c | GSM8K | HumanEval |
|---|---|---|---|
| Weight (Equation (31)) | 38.0 | 25.9 | 18.9 |
| **Activation** (Equation (32)) | **40.0** | **58.2** | **30.5** |
| Gradient Saliency (Equation (34)) | 39.0 | 51.7 | 26.8 |
| Weight×Activation (Equation (33)) | 38.2 | 33.7 | 28.7 |
| Activation×Gradient (Equation (35)) | 38.7 | 46.3 | 26.2 |
| SNIP First-order Sensitivity (Weight×Gradient) (Equation (36)) | 39.2 | 41.5 | 24.4 |

As shown in Table 13, the choice of channel scores has a non-trivial impact on performance: results on the commonsense task (ARC-c) varies moderately across metrics, whereas the gap becomes substantial on reasoning-heavy benchmarks (GSM8K and HumanEval). For example, on GSM8K, *Activation* reaches 58.2, while *Weight* drops to 25.9, and other weight or gradient based variants also lag behind. Overall, we use *Activation* as the default channel scoring metric in all experiments.

### C.3.2. FIRST-ORDER VS. SECOND-ORDER ATTRIBUTION SCORE

Our default attribution-based score $s_e^{(1)}$ is a first-order Taylor approximation, motivated by computational efficiency. To verify that this approximation does not compromise allocation quality, we additionally implement a lightweight but *exact* second-order proxy $s_e^{(2)}$ by perturbing each expert output with a scalar $\alpha_e$ and benchmark both against the true ablated score $s_e^{(\text{true})} = \mathcal{L}_e(0) - \mathcal{L}_e(1)$. Table 14 shows that $s_e^{(1)}$ already correlates highly with $s_e^{(\text{true})}$ (Pearson 0.959; channel-allocation Pearson 0.966), while $s_e^{(2)}$ matches it almost exactly. The second-order proxy improves the end-to-end average by +1.2%, at the cost of $\sim$17× longer calibration. The first-order score thus remains a strong default, and the second-order proxy serves as an enhanced variant whenever the additional calibration budget is acceptable.

*Table 14.* Comparison of the first-order attribution score $s_e^{(1)}$ used in our main results and an exact second-order proxy $s_e^{(2)}$ on Qwen1.5-MoE-A2.7B under $\mathbf{P}_{25\%}$ $\mathbf{Q}_{4b}$. The second-order proxy is also benchmarked against the true ablated score $s_e^{(\text{true})}$ computed by directly removing each expert.

| Comparison | Pearson ↑ | L1 ↓ | Channel Pearson ↑ | Channel Diff ↓ |
|---|---|---|---|---|
| $s_e^{(1)}$ vs. $s_e^{(\text{true})}$ | 0.959 | 0.058 | 0.966 | 23.4 |
| $s_e^{(2)}$ vs. $s_e^{(\text{true})}$ | 1.000 | 0.000 | 1.000 | 0.13 |

| Score | PIQA | ARC-c | ARC-e | BoolQ | HellaSwag | WinoGrande | GSM8K | MMLU | Avg |
|---|---|---|---|---|---|---|---|---|---|
| First-order $s_e^{(1)}$ (default) | 74.45 | 35.93 | **66.07** | 70.46 | 61.28 | 69.26 | **58.24** | 51.43 | 60.89 |
| Second-order $s_e^{(2)}$ | **78.04** | **36.13** | 64.67 | **76.45** | **61.30** | **70.26** | 56.49 | **53.40** | **62.09** |

### C.3.3. Loss Smoothing

**Raw Loss vs. Smoothed Losses as the Target Coverage Ratio**   Figure 11 illustrates how the square-root smoothing transforms the raw layerwise loss into a more balanced coverage target, and how that target translates into the final channel keep ratio.

The top colorbar shows the raw ablated loss per layer, which spans a wide dynamic range: a small number of critical layers dominate the signal while most layers contribute only modestly. Directly using raw loss as the inter-layer importance signal would therefore concentrate the retained budget on a few layers and drastically under-budget the rest. After applying square-root smoothing (middle colorbar), the dynamic range is compressed: the most sensitive layers are down-weighted, and moderately sensitive layers receive a proportionally larger share of the budget. The resulting score-coverage targets are more uniformly distributed across layers, enabling a stable and globally balanced pruning allocation.

The bottom colorbar shows the actual channel keep ratio produced by the coverage-maximized allocation under this smoothed target. Layers that are assigned a higher coverage ratio (darker color) retain more of their channels, while layers with highly concentrated scores can meet the same target with a smaller fraction. Comparing the middle and bottom colorbars illustrates the decoupling between coverage target and channel count that is central to our method: a high coverage target does not imply a large channel budget when the score distribution is concentrated.

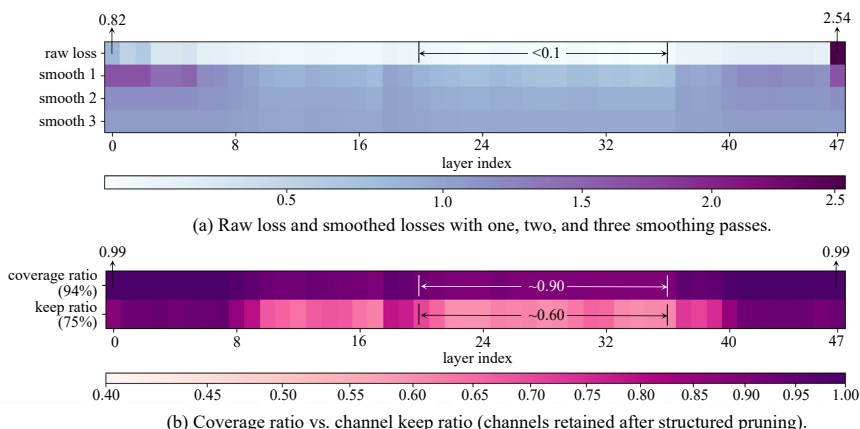

(a) Raw loss and smoothed losses with one, two, and three smoothing passes.

(b) Coverage ratio vs. channel keep ratio (channels retained after structured pruning).

*Figure 11.* Losses (raw and smoothed), coverage ratio and channel keep ratio after pruning.

**Alternative Smoothing Functions for Layerwise Loss**   The square-root smoothing used in inter-layer allocation is not a theoretically essential component; rather, it is a simple realization of monotone-concave dynamic-range compression. Without smoothing, a few high-loss layers capture most of the channel budget while low-loss layers are over-pruned. Applying any monotone-concave transform reduces this imbalance by suppressing outlier values while preserving the relative ordering of layers.

We compare the default $\sqrt{\cdot}$ smoothing against three standard alternatives. Let $x \geq 0$ denote the raw layerwise loss, $\mu$ and $\sigma$ the mean and standard deviation of the losses across all layers.

- **Square-root (ours):** $g(x) = \sqrt{x}$.

- **Log smoothing:** $g(x) = \log(1 + \alpha x)$,    with $\alpha = 5$.

- **Huber-style smoothing:**

$$g(x) = \begin{cases} x, & x \leq \delta, \\ \delta + \sqrt{\delta\,(x - \delta)}, & x > \delta, \end{cases} \qquad \delta = \mu + 0.5\,\sigma.$$

- **Clip-based smoothing:** $g(x) = \mathrm{clip}(x,\ \mu - k\sigma,\ \mu + k\sigma)$,    with $k = 0.5$.

All four functions are monotone (preserving relative layer ordering) and concave (compressing the dynamic range). All smoothed variants substantially outperform the unsmoothed baseline as shown in Table 15, supporting the need for dynamic-range compression. The square-root transform gives the best result while requiring no hyperparameter tuning.

*Table 15.* Downstream accuracy with channel allocations derived from different monotone-concave smoothing functions of the layerwise loss on Qwen1.5-MoE-A2.7B under $\mathbf{P}_{50\%}$.

| Smooth fn. | Setting | PIQA | ARC-c | ARC-e | BoolQ | HellaSwag | WinoGrande | GSM8K | HumanEval | Avg |
|---|---|---|---|---|---|---|---|---|---|---|
| None | – | 72.06 | 34.73 | 54.29 | 70.46 | 58.48 | 64.07 | 46.82 | 18.29 | 52.40 |
| Log | $g(x)=\log(1+5x)$ | 74.85 | 32.34 | 60.88 | 68.66 | 61.07 | 68.86 | 56.40 | 32.32 | 56.92 |
| Huber-style | $\delta=\mu+0.5\sigma$ | 73.45 | 32.14 | 56.89 | 67.66 | 61.65 | 69.86 | 56.40 | 29.27 | 55.92 |
| Clip | $k=0.5$ | 74.45 | 35.33 | 59.88 | 68.66 | 62.08 | 68.66 | 55.80 | 28.66 | 56.69 |
| **Sqrt (ours)** | **–** | **74.45** | **35.93** | **66.07** | **70.46** | **61.28** | **69.26** | **58.24** | **30.50** | **58.27** |

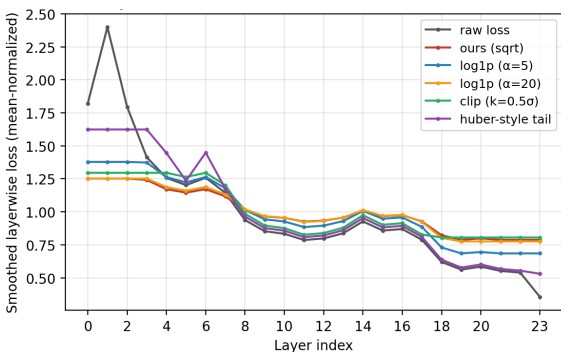

*Figure 12.* Smoothed layerwise loss under different monotone-concave smoothing functions.

### C.3.4. HYPERPARAMETER SENSITIVITY IN CBA AND AAR

For Coverage-Maximized Budget Allocation, we set the maximum number of binary-search iterations to 50. In practice, the search usually converges within 30 iterations, and the maximum only serves as a safeguard. For Alignment-Aware Redistribution, the minimum kept-channel threshold $m$ prevents overly thin experts and is constrained by hardware block size. We ablate $m \in \{64, 128, 256, 512\}$ in Table 16; $m = 128$ provides the best trade-off and is used by default. The residual reallocation strategy is summarized in the main text in Table 8, with full per-task results in Table 17.

### C.4. Sensitivity and Robustness Analysis

#### C.4.1. SENSITIVITY TO CALIBRATION CORPUS

Our default setup follows common post-training compression practice, using C4 for general tasks, GSM8K for math, and OpenCodeReasoning for code. To examine the sensitivity systematically, we conduct an ablation on six calibration corpora: WikiText2, C4, Pile, RedPajama, GSM8K (train), and OpenCodeReasoning. Results in Table 18 show that general tasks are relatively robust to general-domain corpora, while domain-specific tasks benefit from domain-matched calibration. The sensitivity is therefore structured rather than arbitrary.

#### C.4.2. ROBUSTNESS ACROSS ROUTING POLICIES

Our method is not tied to a specific routing design. The main-text experiments cover Qwen-style standard top-$k$ routing and DeepSeek-style routing with load-balancing considerations. Figure 13 visualizes router entropy across tasks, layer depths, and MoE models, showing that the evaluated settings cover different routing dynamics rather than a single homogeneous pattern.

Figure 14 further reports expert activation magnitudes across representative shallow, middle, and deep layers of Qwen1.5-MoE-A2.7B under different calibration corpora. The distributions differ across both layers and corpora, confirming that expert heterogeneity is not an artifact of a single calibration source.

To further evaluate robustness under different routing budgets, we switch the activated experts from top-2 to top-1 on Qwen1.5-MoE-A2.7B and DeepSeek-V2-Lite. As expected, accuracy drops when fewer experts are activated, but our method still preserves most of the original performance at 50% pruning under both top-1 and top-2 routing.

*Table 16.* Ablation on the minimum-channel threshold $m$ in AAR on Qwen1.5-MoE-A2.7B under $\mathbf{P}_{25\%}$ $\mathbf{Q}_{4b}$.

| $m$ | PIQA | ARC-c | ARC-e | BoolQ | HellaSwag | GSM8K | HumanEval | WinoGrande | Avg |
|---|---|---|---|---|---|---|---|---|---|
| 64 | 78.44 | 40.52 | 63.67 | 77.25 | 63.17 | 54.29 | 27.44 | 67.66 | 59.06 |
| **128** | **78.72** | **41.51** | **63.48** | **77.12** | **63.62** | **56.88** | **26.83** | **68.16** | **59.54** |
| 256 | 78.64 | 41.32 | 63.87 | 77.25 | 63.47 | 56.69 | 26.83 | 68.26 | 59.54 |
| 512 | 79.04 | 39.72 | 63.07 | 77.45 | 63.67 | 55.49 | 26.22 | 68.46 | 59.14 |

*Table 17.* Comparison of two AAR residual reallocation strategies on Qwen1.5-MoE-A2.7B with different alignment block sizes $a$. *l-r-c*: *largest removed channels*, *l-r-s*: *largest removed scores*.

| Strategy | $a$ | GSM8K | HumanEval | PIQA | ARC-c | ARC-e | BoolQ | HellaSwag | WinoGrande | Avg |
|---|---|---|---|---|---|---|---|---|---|---|
| l-r-c | 64 | 56.29 | 26.83 | 78.44 | 40.52 | 63.67 | 77.25 | 63.17 | 67.66 | 59.23 |
| l-r-c | 128 | 56.29 | 27.44 | 78.72 | 41.51 | 63.48 | 77.12 | 63.62 | 68.16 | 59.54 |
| l-r-c | 256 | 57.29 | 24.39 | 78.64 | 41.32 | 63.87 | 77.25 | 63.47 | 68.26 | 59.31 |
| l-r-s | 64 | 54.69 | 28.05 | 78.04 | 40.72 | 63.67 | 77.45 | 63.47 | 67.47 | 59.20 |
| l-r-s | 128 | 55.89 | 26.83 | 78.64 | 39.92 | 61.88 | 76.85 | 63.27 | 69.66 | 59.12 |
| l-r-s | 256 | 56.69 | 24.39 | 79.64 | 41.52 | 66.07 | 75.65 | 64.07 | 71.46 | 59.94 |

## C.5. Visualizations

### C.5.1. VISUALIZATION OF LOSS, SCORES AND SPARSITY ALLOCATION AT EXPERT-LEVEL

Figures 15 and 16 visualize the expert-level channel-score distribution and the resulting allocation for Qwen1.5-MoE-A2.7B and Qwen3-30B-A3B, respectively. Each bar stack represents one expert: darker segments at the top correspond to high-scoring channels, while lighter segments reflect channels with smaller scores. The fraction of dark segments therefore indicates how concentrated an expert's score is. Experts whose information is packed into a small number of channels exhibit darker, narrower stacks.

The red lines report the fraction of channels retained after coverage-maximized allocation. Experts with a highly concentrated score distribution (tall dark segments, small light tails) are assigned a smaller channel budget, because a high coverage target can be met by keeping only the top channels. Conversely, experts with flat, spread-out distributions require a larger kept-channel fraction to reach the same coverage threshold. The yellow diamond markers show the attribution score of each expert, which controls the per-expert coverage target in our intra-layer allocation. Experts with higher attribution scores receive a tighter coverage target (more channels retained) to avoid degrading high-contribution experts, whereas low-attribution experts are compressed more aggressively.

For Qwen3-30B-A3B, Figure 16 further compares two channel-score metrics: activation-based scores (panels a, b) and weight-gradient-based scores (panels c, d). Both metrics lead to similar kept-channel allocations (red lines), confirming that the coverage-maximized allocation is robust to the choice of scoring metric.

## D. More Related Works

**Efficient MoE.** MoE compression and acceleration have attracted increasing interest as models continue to grow in scale, from Mixtral $8\times7$B with 8 activated experts (Jiang et al., 2024), to Qwen3-235B-A22B with 128 experts among which 8 are activated for each token (Qwen-Team, 2025).

Existing methods can be broadly categorized into four major techniques. (1) **Expert trimming** removes a subset of experts through data driven selection, so that low contribution experts are never loaded or computed, reducing both memory footprint and computations (Liu et al., 2024a; Bai et al., 2025; Muzio et al., 2024; Chowdhury et al., 2024; Dong et al., 2025; Lu et al., 2024; Zhang et al., 2025). (2) **Expert skipping** is a complementary approach that retains the full expert pool while skipping the computation of low importance experts at inference time, typically through routing thresholds or dynamic gating (Liu et al., 2024a; Bai et al., 2025; Lu et al., 2024; ?; Chen et al., 2025a). (3) **Expert slimming** compresses the internal structure and parameter of each expert by pruning, quantization, or low rank decomposition, while keeping the number of experts fixed (Yang et al., 2024; He et al., 2025; Lee et al., 2025; Xie et al., 2024; ?; Chen et al., 2025b;a). (4) **Expert merging** clusters experts with similar behavior or activation patterns and combines them into fewer experts by averaging, SVD based factorization, or pairwise merging strategies (Zhang et al., 2024; Li et al., 2025; Zhao et al., 2025; ?).

Despite the diverse aspects, most existing works only rank experts at the granularity of entire experts and do not explicitly

*Table 18.* Sensitivity of Ours$_Q$ to the choice of calibration corpus on Qwen1.5-MoE-A2.7B under $\mathbf{P}_{25\%}$ $\mathbf{Q}_{4b}$.

| Calibration data | HellaSwag | WinoGrande | PIQA | ARC-c | MMLU | GSM8K | HumanEval |
|---|---|---|---|---|---|---|---|
| WikiText2 | 57.83 | **70.26** | 74.85 | 36.11 | 51.59 | 27.15 | 2.44 |
| C4 | **61.28** | 69.26 | 74.45 | 35.94 | 52.46 | 9.18 | 3.05 |
| Pile | 58.68 | 65.27 | 75.65 | **39.92** | **54.18** | 42.51 | 16.46 |
| GSM8K (train) | 61.22 | 65.45 | **76.05** | 37.13 | 49.35 | **61.08** | 15.85 |
| OpenCodeReasoning | 58.25 | 62.28 | 74.85 | 33.95 | 52.56 | 54.49 | **29.27** |
| RedPajama | 60.61 | 65.88 | 73.65 | 36.51 | 52.43 | 23.35 | 3.05 |

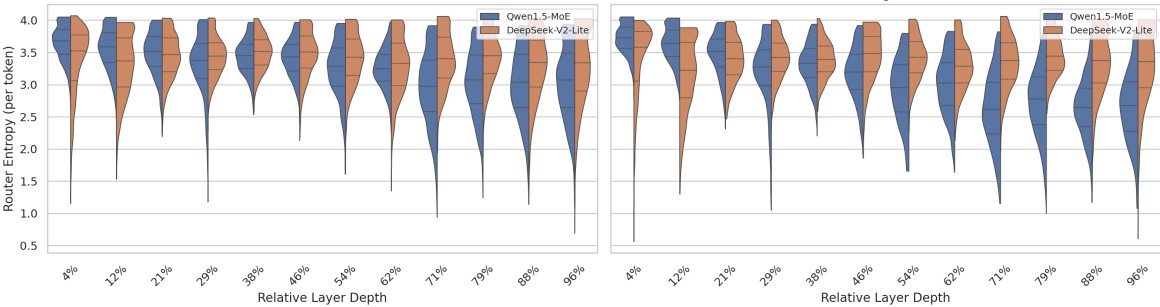

*Figure 13.* Router entropy distributions across tasks, layer depths, and MoE models evaluated in the main text. The distributions illustrate the routing-dynamics variation used to evaluate robustness across architectures and tasks.

analysis the redundancy within each expert. MoE-I$^2$ (Yang et al., 2024) reduces the parameters via low rank decomposition and assigns higher ranks to more important experts while using lower ranks for less important ones. However, the speedup is limited: the fragmentation into small kernels makes it difficult to reach peak throughput of one larger kernel, introducing additional overhead in kernel launching, cache hit, and memory access. Chen et al. (Chen et al., 2025b) quantize all parameters to low bitwidth, compare the reconstruction error, and assign higher bitwidth to experts that are more sensitive to quantization. This strategy, however, only feasible to methods that have a small search space, e.g., 4/8 bitwidth, which is insufficient for fine grained expert-wise compression budget allocation.

**Expert Importance.** A key driver behind MoE efficiency designs is the highly unbalanced contribution of different experts. This has motivated a variety of methods for measuring expert importance. (1) **Router based** statistics are widely used, including average gate scores, processed token counts (expert hit rates), and gate variations during fine tuning (He et al., 2025; Lee et al., 2025; Xie et al., 2024; Muzio et al., 2024; Chowdhury et al., 2024; Dong et al., 2025; Li et al., 2025; Lu et al., 2024; ?; Chen et al., 2025b). (2) **Activation based** metrics such as gate weighted outputs or activation saliency are also employed (Dong et al., 2025; Li et al., 2025; Zhang et al., 2025; Zhao et al., 2025). (3) **Loss or accuracy based** criteria measure the performance drop when removing a particular expert or a subset of experts. For example, they quantify the impact on reconstruction loss or downstream task performance after compression (Liu et al., 2024a; Yang et al., 2024; Zhang et al., 2024; Lu et al., 2024). (4) **Learnable method** is recently proposed, which learns a set of importance scalars that are jointly optimized during fine tuning (Bai et al., 2025).

However, one limitation is that, router based and performance based statistics are often *not comparable across layers*. Routers and activations in different layers may behave very different in decision patterns, or follow different distributions. Loss values can be depth dependent and unstable under different experimental setups, including the source of calibration data, the loss function, and the tokenization scheme. As a result, many previous methods resort to assigning a uniform compression ratio to all layers instead of performing cross layer importance comparison.

A second limitation is that most existing metrics exhibit *a high dynamic range* and are primarily used to rank experts and entirely trim the least important $k$ experts, rather than to support precise allocation of compression ratios. Only a few works explore redundancy within experts. MoE-I$^2$ and Liu et al. (Liu et al., 2024a) remove a small group of experts at a time and compare the resulting loss increase or accuracy drop in order to infer expert importance within each layer. But such loss or accuracy based methods are only feasible for relatively small MoE models with a limited number of experts. When the search space over layers and experts grows, even greedy or genetic strategies incur prohibitively high computational cost, which severely restricts their applicability to modern large scale MoE architectures.

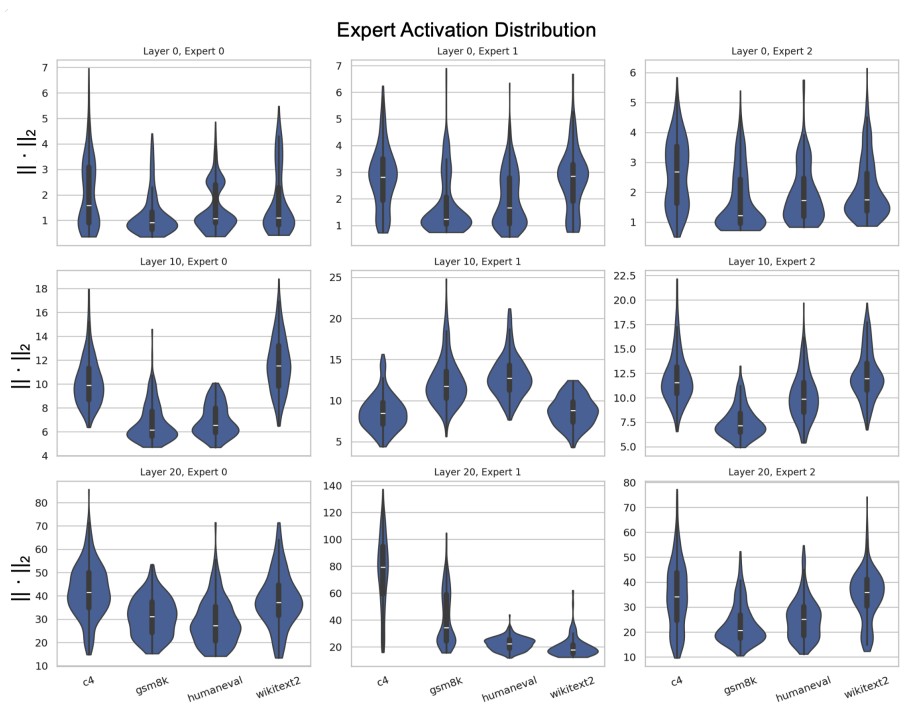

*Figure 14.* Distribution of expert activation magnitudes across representative shallow, middle, and deep layers of Qwen1.5-MoE-A2.7B under different calibration corpora. The figure supports the calibration robustness analysis by showing that activation heterogeneity persists across data sources.

*Table 19.* Evaluation under different top-$k$ routing strategies for Qwen1.5-MoE-A2.7B and DeepSeek-V2-Lite, with and without 50% structural pruning.

| Model | Top-$k$ | Prune (%) | PIQA | ARC-c | ARC-e | BoolQ | HellaSwag | WinoGrande | GSM8K | HumanEval | Avg |
|---|---|---|---|---|---|---|---|---|---|---|---|
| Qwen1.5-MoE-A2.7B | 4 | 0 | 80.79 | 40.41 | 69.44 | 70.57 | 77.17 | 69.77 | 61.50 | 34.20 | 62.98 |
| | 4 | 50 | 74.45 | 35.93 | 66.07 | 70.46 | 61.28 | 69.26 | 58.24 | 30.50 | 58.27 |
| | 2 | 0 | 83.03 | 42.12 | 65.27 | 77.45 | 64.87 | 65.07 | 52.50 | 34.15 | 60.56 |
| | 2 | 50 | 72.65 | 35.13 | 63.27 | 66.67 | 60.28 | 67.66 | 50.50 | 25.00 | 55.15 |
| | 1 | 0 | 79.24 | 40.12 | 66.87 | 69.26 | 61.08 | 65.07 | 36.93 | 20.12 | 54.84 |
| | 1 | 50 | 71.86 | 33.53 | 59.28 | 66.07 | 56.09 | 63.07 | 29.74 | 21.34 | 50.12 |
| DeepSeek-V2-Lite | 6 | 0 | 80.20 | 46.93 | 78.37 | 79.82 | 77.98 | 71.35 | 30.90 | 32.30 | 62.23 |
| | 6 | 50 | 78.40 | 42.49 | 73.78 | 71.77 | 74.59 | 69.22 | 33.70 | 28.70 | 59.08 |
| | 2 | 0 | 79.45 | 42.66 | 68.30 | 75.15 | 62.82 | 65.56 | 25.64 | 25.61 | 55.65 |
| | 2 | 50 | 75.34 | 35.42 | 61.25 | 65.36 | 57.34 | 63.01 | 24.85 | 18.29 | 50.11 |
| | 1 | 0 | 73.65 | 33.53 | 61.88 | 65.07 | 54.69 | 60.68 | 6.39 | 10.98 | 45.86 |
| | 1 | 50 | 71.06 | 29.74 | 56.49 | 61.48 | 52.89 | 57.29 | 8.38 | 7.93 | 43.16 |

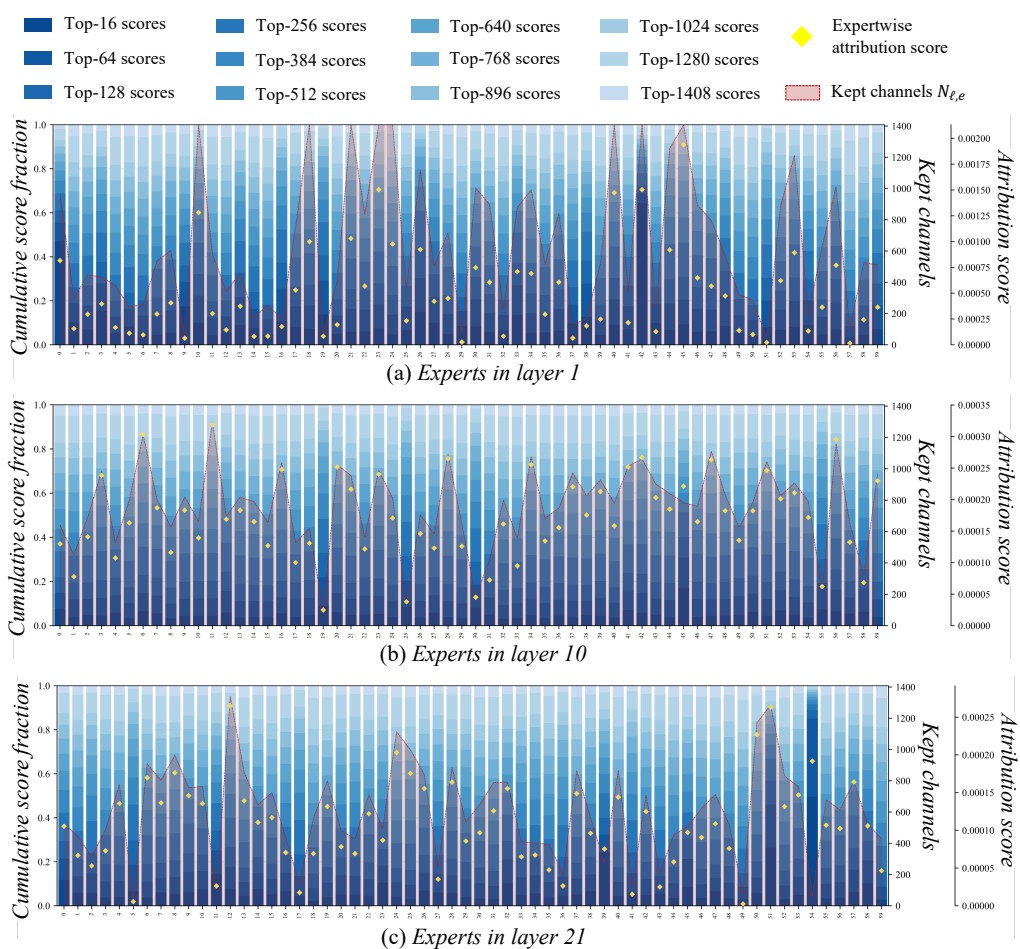

(a) *Experts in layer 1*

(b) *Experts in layer 10*

(c) *Experts in layer 21*

*Figure 15.* Cumulative scores fraction (blue stacked bars), kepted channels (red lines) and attribution score (yellow diamond markers) for each expert in specific layers in Qwen1.5-MoE-A2.7B.

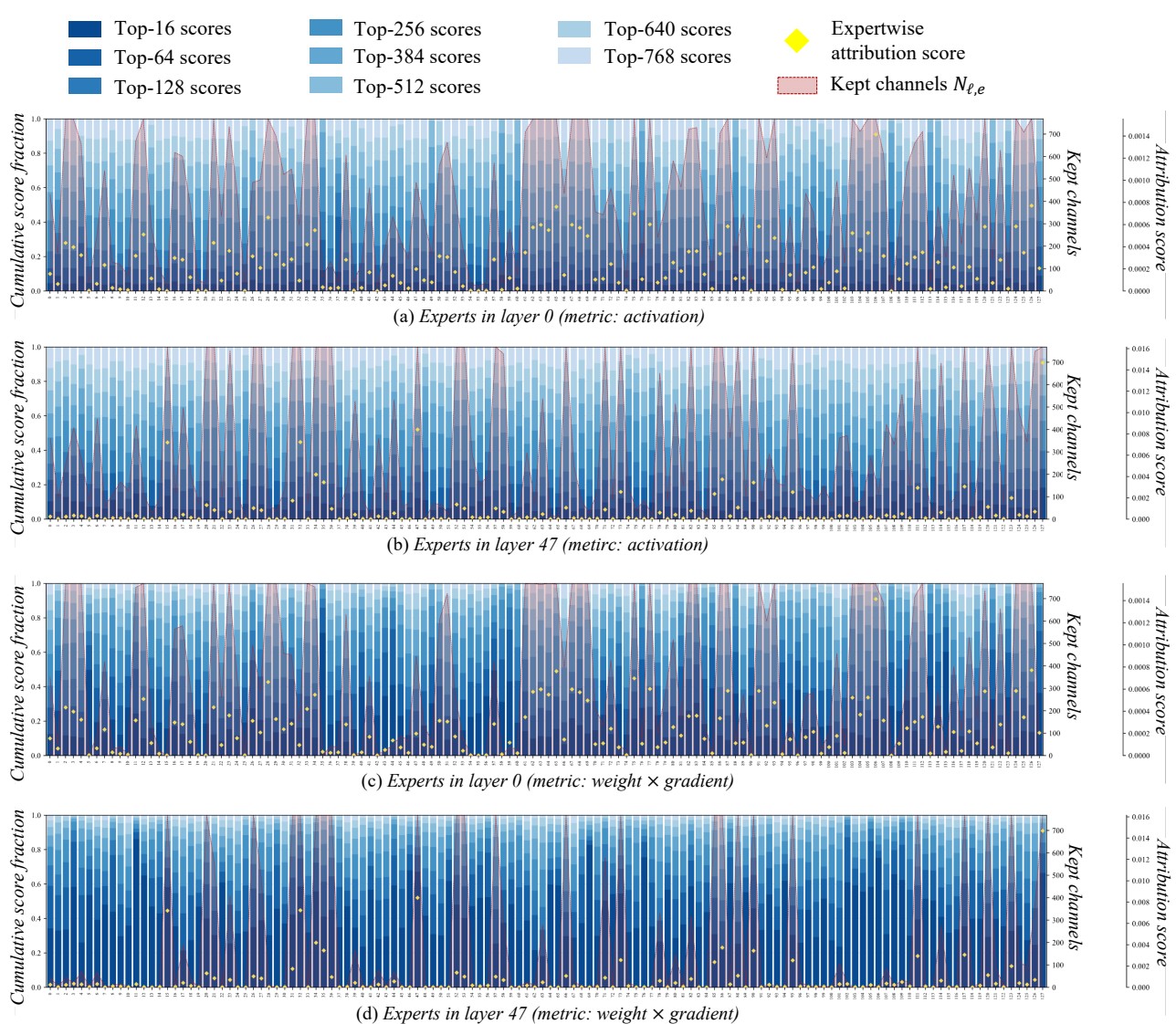

(a) *Experts in layer 0 (metric: activation)*

(b) *Experts in layer 47 (metirc: activation)*

(c) *Experts in layer 0 (metric: weight × gradient)*

(d) *Experts in layer 47 (metric: weight × gradient)*

*Figure 16.* Cumulative channel-score fractions (blue stacked bars), kept channels (red lines), and attribution scores (yellow diamonds) for experts within a representative layer of Qwen3-30B-A3B. Panels (a) and (b) use activation-based scores, while (c) and (d) use weight gradient scores (weight times gradient). The channel-score concentration pattern appears under both metrics. Although experts exhibit substantial heterogeneity, our coverage-maximized budget allocation yields similar kept-channel allocations across metrics (red).

