# OpenReview forum: "Attribution-Guided and Coverage-Maximized Pruning for Structural MoE Compression"
_ICML.cc/2026/Conference — ICML 2026 spotlight_

### Official Review · Reviewer_zn3k · 2026-02-27

**Soundness:** 2
**Presentation:** 2
**Significance:** 3
**Originality:** 3
**Overall Recommendation:** 5
**Confidence:** 4

**Summary:**

This paper proposes an end-to-end expert-wise pruning framework for Mixture-of-Experts (MoE) models, which consists of three core components: Attribution-Guided Loss Approximation (ALA), Coverage-Maximized Budget Allocation (CBA), and Alignment-Aware Redistribution (AAR). The framework reformulates pruning as maximizing channel score coverage to better capture redundancy in MoEs, and incorporates hardware alignment to ensure compatibility with low-bit GEMM kernels. Extensive experiments on mainstream MoE models (e.g., Qwen, DeepSeek) demonstrate that the method achieves significant storage compression and inference speedup while maintaining or even improving model performance on various tasks.

**Compliance With Llm Reviewing Policy:**

Affirmed.

**Key Questions For Authors:**

1.Global Scaling Factor Design. For inter-layer pruning, you use a single global scaling factor α to scale all layers in the CBA algorithm. Why is this a reasonable assumption given the inherent heterogeneity of the model (i.e., significant differences in importance and redundancy across layers)?
2.Explicit Interpretation of the Gray Curve in Figure 5. The core overview figure (Figure 5) lacks a clear legend and annotation for the gray curve. What is the exact physical meaning of this gray curve? If it represents a fixed pruning ratio, why does it show a polygonal line trend? If it denotes another evaluation metric, please explicitly define it and explain the reason for the contradiction between its trend and the proposed formula.
3.Selection of Hyperparameter m. In Section 4.3, you involve the hyperparameter m in the AAR algorithm. What is the setting principle of m, and how do you choose this hyperparameter?
4.ALA Approximation and Gating Value Ignorance. Your ALA method (Section 4.1) ignores the influence of MoE gating values when calculating loss changes via first-order Taylor expansion. Have you conducted quantitative experiments to verify the impact of this omission on expert importance estimation accuracy and final pruning performance? Have you validated the robustness of this approximation across different MoE architectures (e.g., Top-1/Top-2 routing, different gating distribution characteristics)? Can you provide the theoretical proof basis?
5.Selection Strategy of Residual Block Reallocation in AAR Algorithm. In the AAR algorithm, the strategy of prioritizing experts with larger remainders for residual block reallocation is adopted. What theoretical support does this selection strategy have, and is there any formula proof to ensure that it is the optimal strategy for residual block reallocation? Have you also conducted experiments on other allocation strategies, such as prioritizing more important experts?

**Limitations:**

No. The authors do not fully discuss the limitations of their research work or potential negative social impacts.
It is suggested that the authors clarify the limitations of the proposed method, including the sensitivity of the ALA approximation under different MoE routing mechanisms and the generalization ability of the hardware block size parameter across different platforms.

**Strengths And Weaknesses:**

Strengths.In terms of Soundness, this study demonstrates solid technical reliability with a well-designed and fully justified empirical setup. Extensive experiments on multiple large-scale MoE models of varying sizes strongly support the core claim that the proposed method achieves substantial storage reduction and inference acceleration while maintaining or even improving model performance. In-depth analysis and visualization of the channel importance distribution further validate the rationality of the method, forming a complete logical closed loop between theoretical derivation and experimental verification. Regarding Presentation, the study clearly conveys its core contributions and methodological framework overall. The design rationale and technical pathway of the core modules are elaborated in a coherent manner, enabling readers in the field to quickly grasp the research's core value and technical highlights. For Significance, the research precisely addresses the key pain points in the practical deployment of MoE models. The proposed hardware alignment module directly resolves the engineering challenge of incompatibility between compressed models and low-bit GEMM kernels, effectively filling the gap between MoE algorithm optimization and hardware implementation. It boasts high engineering application value, providing a feasible technical solution for the actual deployment of MoE models and driving the engineering development of related fields. In terms of Originality, a novel end-to-end pruning framework for MoE models is proposed, which innovatively integrates three core modules: attribution-guided expert importance estimation, coverage-maximized budget allocation, and hardware-aware channel redistribution. This framework breaks the limitation of traditional MoE pruning that only focuses on algorithm optimization while ignoring hardware constraints. Meanwhile, the study reveals the channel redundancy characteristics of MoE architectures through experiments and visualization, offering new research perspectives and insights for subsequent studies on MoE efficiency improvement.

Weaknesses. In terms of Soundness, this study has several notable deficiencies, as the theoretical rigor and design rationality of the core methods lack sufficient verification. First, the Attribution-Guided Loss Approximation (ALA) method proposed in Section 4.1 fails to account for the inherent influence of gating values in MoE models when calculating the loss change after expert removal via first-order Taylor expansion. No quantitative experiments are conducted to explicitly prove that this omission has a negligible impact on the results of expert importance estimation and the final pruning performance, nor is the robustness of this approximation method verified across different MoE architectures (e.g., Top-1/Top-2 routing), leading to insufficient empirical support for the rationality of its theoretical foundation and leaving inadequate evidence to confirm that it is the optimal method for evaluating expert importance.
Second, the key hyperparameters of the core algorithms lack clear values and setting rationale. For hyperparameters such as the maximum number of iterations for the Coverage-Maximized Budget Allocation (CBA) and the minimal channel threshold m for AAR redistribution, no specific values are provided in either the main text or the appendix, nor are the experimental basis or empirical criteria for parameter setting explained. Third, the strategy of prioritizing experts with larger remainders in the residual block reallocation of the AAR algorithm is not supported by theoretical analysis, and there is no guarantee that this approach is optimal for channel reallocation. Regarding Presentation, the core visualization chart has flaws in information transmission. Figure 5, the key overview chart of the method, lacks complete legend annotations and detailed captions. It does not clarify the specific physical meaning of the gray curve, nor does it provide any specific explanation for it. The ambiguity of key visual information greatly increases the difficulty for readers to understand the core logic of the method and prevents the intuitive verification of the method's design rationality through the chart. For Significance, the research accurately targets the core pain points in the engineering deployment of MoE models, and the proposed method exhibits distinct engineering application value and field contributions in terms of storage compression, inference acceleration, and performance preservation, with no obvious core flaws. The existing deficiencies are only concentrated in the perfection of technical details and the standardization of expression, which do not have a substantive impact on the core value and practical application significance of the research. In terms of Originality, although the overall framework demonstrates significant novelty, some core design choices (e.g., the single global scaling factor α) are overly simplified, and no ablation experiments are conducted to compare its performance with that of layer/expert-specific local scaling factors, making it impossible to prove the optimality of this design. This somewhat weakens the presentation of the research's innovativeness and fails to fully highlight the ingenuity and innovative points of the method design, yet it does not negate the novelty and core contributions of the overall method.

---

> ### Author Rebuttal · Authors · 2026-03-30
>
> We thank Reviewer zn3k for the constructive feedback.
>
> Additional derivation, Fig. R1, Tab. R4-R6: [For-Reviewer-zn3k.md](https://anonymous.4open.science/r/moe-prune/additional/For-Reviewer-zn3k.md)
>
>
> **[W1/Q4] ALA ignores gating values in the first-order**
>
> We clarify that ALA does not ignore gating values. The gate is already absorbed into the gradient term through the chain rule. Specifically,
>
> $$
> \frac{\partial \mathcal{L}}{\partial z_{\ell,e}} = \left(\frac{\partial y_\ell}{\partial z_{\ell,e}}\right)^\top \frac{\partial \mathcal{L}}{\partial y_\ell} = g_{\ell,e}(h_\ell) \frac{\partial \mathcal{L}}{\partial y_\ell}
> $$
>
> Substituting this into the first-order loss change yields
>
> $$
> \Delta \mathcal{L}^{(e)}_ \ell \approx - \frac{ \partial \mathcal{L} } {\partial z_{\ell,e}} z_{\ell,e}(h_\ell)
> $$
>
> Detailed derivation is provided in For-Reviewer-zn3k.md and will be added in the revision.
>
>
>
> **[W2/Q5] Robustness across routing architectures**
>
> Our method is not tied to a specific routing design.
>
> **Router dynamics.** The current experiments already cover Qwen-style standard top-k routing, and more constrained routing of DeepSeek with load balancing considerations. We visualize the router statistics for Qwen and DS MoEs in Fig. R1 to show the dynamics.
>
> **Top-1 / Top-2 routing.** Results are listed in Tab. R1, with per-task results in Tab. R4. Accuracy drops when fewer experts are activated, as expected, but our method still preserves most of the original performance.
>
> |Model|Top-k|P%|Avg|
> |-|-|-|-|
> |Qwen1.5-MoE-A2.7B|2|0%|62.98|
> ||2|50%|58.27|
> ||1|0%|60.56|
> ||1|50%|55.15|
> |DS-V2-Lite|2|0%|54.84|
> ||2|50%|50.12|
> ||1|0%|62.23|
> ||1|50%|59.08|
>
> > Tab. R1
>
>
>
> **[W3/Q3] Missing key hyperparameters**
>
> We agree this should be stated more clearly.
>
> For **CBA**, we set the maximum iterations to 50. In practice, the algorithm usually converges within 30 iterations, and the maximum only serves as a safeguard. For the stopping tolerance, we use
>
> $$
> \left|N(\rho)-N_{\text{budget}}\right|<\varepsilon N
> $$
>
> with $\varepsilon=0.2\%$, which usually corresponds to only 1 to 2 channels and has negligible impact in practice.
>
> For **AAR**, the minimum channel is **$m=128$**. It avoids overly thin experts and is also constrained by hardware block size. We provide an ablation on $m$ below on Qwen MoE, with per-task results in Tab. R5.
>
> |$m$|Avg|
> |-|-|
> |64|59.06|
> |**128**|**59.54**|
> |256|59.54|
> |512|59.14|
>
>
> > Tab. R2
>
> Too small $m$ keeps weak thin experts, while too large $m$ removes useful experts. A moderate threshold works better. We will add the values and roles of these hyperparameters in the revision.
>
>
>
> **[W4/Q5] AAR residual reallocation strategy**
>
> We agree that the current largest-remainder rule is a heuristic.
>
> **Design rationale.** AAR aims to make the final block-aligned allocation as close as possible to the pre-alignment CBA solution by compensating experts that lost more channels.
>
> **Alternative strategy comparison.** We further design an importance-aware variant that prioritizes *largest removed scores* (l-r-s), comparing with the current *largest removed channels* strategy (l-r-c). Results are shown in Tab. R3, with per-task results in Tab. R6.
>
> |Metric|$a$|Avg|
> |-|-|-|
> |l-r-c|64|59.23|
> |l-r-c|128|59.54|
> |l-r-c|256|59.31|
> |l-r-s|64|59.2|
> |l-r-s|128|59.12|
> |l-r-s|256|59.94|
>
> > Tab. R3
>
> The two strategies are very close, suggesting low sensitivity to the sorting criterion. We will include the alternative strategy in comparison.
>
>
>
> **[W5/Q2] Figure 5 unclear**
>
> We apologize for the unambiguous figure. The gray curve in Figure 5 represents a naive baseline and has no additional meaning. We will revise the figure.
>
>
> **[W6/Q1] Overly simplified scaling factor $\alpha$**
>
> We clarify that the method does not use one single global $\alpha$ to uniformly scale all layers and experts.
>
> The confusion comes from our condensed presentation. In fact, $\alpha$ is a **search anchor** used in two separate stages:
>
> - In inter-layer allocation, $\alpha$ is used together with layerwise loss to determine **layer-specific** pruning ratios.
> - In intra-layer allocation, each layer has its own search anchor, $\alpha_l$, combined with expert-wise ALA scores to determine **expert-specific** retained channels.
>
> Therefore, the final allocation is both layer-specific and expert-specific. We will rewrite this part to clarify the roles of $\alpha$ and $\alpha_l$.
>
>
>
> **[L1] Generalization across platforms**
>
> The reviewer's point is well taken, and we agree this aspect deserves explicit discussion. The block size is not a universal constant, but a hardware-aware parameter determined by kernel alignment, tile size, vectorized memory access, and matrix multiplication design on the target platform.
>
> Accordingly, the key idea of AAR is not a fixed block size, but reallocating arbitrary pruned channels into **hardware-friendly block-aligned shapes**. This principle is transferable.

---

> > ### Author Rebuttal · Reviewer_zn3k · 2026-04-02
> >
> > The author has fully cleared up my questions and doubts.

---

### Official Review · Reviewer_fDBp · 2026-03-03

**Soundness:** 3
**Presentation:** 4
**Significance:** 3
**Originality:** 4
**Overall Recommendation:** 5
**Confidence:** 4

**Summary:**

The authors propose a channel-level pruning framework for MoE models. Compared to traditional expert-level pruning methods, this paper leverages the first-order Taylor expansion to calculate the loss contribution of each expert. By employing a coverage-maximization strategy, dynamically retain high-contribution channels for each expert and further integrate with an alignment-aware redistribution algorithm to optimize hardware kernel boundaries. Empirical results demonstrate that the proposed framework achieves superior compression efficiency compared to most prevalent MoE compression schemes while maintaining near-lossless inference accuracy.

**Compliance With Llm Reviewing Policy:**

Affirmed.

**Final Justification:**

The author addressed my concerns in the rebuttal, so I decided to raise my rating from 4 to 5.

**Key Questions For Authors:**

1. To further justify the reliability of the first-order Taylor approximation, could the authors provide a comparative analysis between the loss increments estimated using the first-order approximation and those incorporating second-order information, to evaluate whether neglecting higher-order terms introduces significant discrepancies?
2. Could the authors clarify the rationale behind the different method selections in Tables 3 and 5? For instance, in Table 5, why is DeepSeek-V2-Lite compared with the MoNE method but not with the others? Is the P38%Q3.03b configuration the standard experimental setup used in the paper? In the comparative study, why were some methods omitted from the quantization evaluation? Are they incompatible?
3. According to my understanding, channel-level approaches such as ALA+CBA operate at a finer granularity than expert-level pruning. In principle, this finer granularity should allow more redundant parameters to be removed while preserving accuracy. However, the experimental results presented in the paper do not seem to clearly substantiate this advantage, which appears somewhat inconsistent with this expectation. Could the authors provide a comparison between ALA+CBA and existing expert-level pruning methods under the same compression ratios (e.g., 25%–75%)?
4. Regarding the P25%Q4b results in Figure 6, is this the global optimum in terms of the accuracy-efficiency trade-off? If possible, could the authors provide a table showing the performance across a wider range of P and Q combinations?

If the author can solve my concerns, I will consider raising my rating.

**Limitations:**

yes

**Strengths And Weaknesses:**

Strengths：
1. The writing is clear and logically coherent, and the figures and tables are well presented and easy to interpret.
2. The first work to explore MoE model compression at the channel level, providing valuable guidance.
3. Comprehensive descriptions of the algorithms and experimental configurations are provided, facilitating reproducibility.
4. The experiments are comprehensive. The proposed method is evaluated against existing compression approaches on multiple mainstream benchmarks and achieves strong performance. The ablation studies further validate the effectiveness of the method.

Weaknesses：
1. The proposed ALA method estimates loss contribution using a first-order Taylor expansion. However, removing an entire expert may introduce large-scale perturbations rather than small local variations. Ignoring second- and higher-order terms could therefore result in substantial approximation errors, which raises my concern.
2. Inconsistency in experimental setup and lack of controlled variables. I observed discrepancies between Table 3 and Table 5, and the methods compared in these tables are not uniform. The pruning ratios seem to vary across different experimental setups without a clear rationale.

---

> ### Author Rebuttal · Authors · 2026-03-30
>
> We thank Reviewer fDBp for the detailed and constructive review.
>
> Additional Fig. R1, Tab. R7-R12, and references: [For-Reviewer-fDBp.md](https://anonymous.4open.science/r/moe-prune/additional/For-Reviewer-fDBp.md)
>
>
>
> **[W1/Q1] First-order Taylor approximation**
>
> In the original submission, we use the first-order because it is efficient and widely used, while exact second-order on full weight matrices is too expensive [R1, R2, R3]. To address this, we introduce a **lightweight but exact second-order proxy** by introducing a scalar perturbation $\alpha_e$:
>
> - $-\frac{\partial \mathcal L_e}{\partial \alpha_e}$: first-order removal gain ($s_e^{(1)}$)
> - $-\frac{\partial \mathcal L_e}{\partial \alpha_e} + \frac{1}{2}\frac{\partial ^2 \mathcal L_e}{\partial \alpha^2_e}$: second-order removal gain ($s_e^{(2)}$)
>
> We compare the three scores in Tab. R1. $s_e^{(\mathrm{true})}$ is the true local perturbation loss.
>
> |Comparison|Pearson ↑|L1 ↓|Channel Pearson ↑|Channel Diff ↓|
> |-|-|-|-|-|
> |$s_e^{(1)}$ vs $s_e^{(2)}$|0.959|0.058|0.966|23.4|
> |$s_e^{(1)}$ vs $s_e^{(\mathrm{true})}$|0.959|0.058|0.966|23.4|
> |$s_e^{(2)}$ vs $s_e^{(\mathrm{true})}$|1.000|0.000|1.000|0.13|
>
> > Tab. R1
>
> Results show that the first-order score is highly correlated with the second-order, and the resulting channel allocation is also very close.
>
> Runtime is reported in Tab. R7. The second-order costs **16.9x** more time than first-order.
>
> The end-to-end downstream performance is shown in Tab. R2, with detailed in Tab. R8.
>
> |Metric|Avg|
> |-|-|
> |$s_e^{(1)}$|60.89|
> |$s_e^{(2)}$|62.09|
>
> > Tab. R2
>
> The second-order proxy improves performance. We will include this as an enhanced variant in the revision.
>
>
>
> **[W2] Inconsistency in settings**
>
> We'd like to clarify that our settings are consistent:
>
> - 50% pruning
> - 25% pruning + 4-bit quantization
>
> Baselines differ in compression techniques, so we report storage size (GB) as a more unified and practical metric.
>
> Methods appear in different tables because we only include results offically reported.
>
> To address the reviewer’s concern, we reproduced MoNE, Unified MoE Compression (He et al., 2025), and C-Prune. Results are shown in Tab. R3, with per-task results in Tab. R9. Results further support the advantage of our method.
>
> |Model|Setting|Method|Avg|
> |-|-|-|-|
> |Qwen1.5-MoE-A2.7B|0%|Baseline|68.32|
> ||50%|MoNE|55.57|
> ||50%|He et al.|47.33|
> ||**50%**|**Ours**|**65.14**|
> ||25%|MoNE|64.65|
> ||25%, 4bit|He et al.|61.96|
> ||**25%, 4bit**|**Ours**|**69.07**|
> |DS-MoE-16B|0%|Baseline|68.60|
> ||50%|MoNE|33.24|
> ||50%|He et al.|44.17|
> ||**50%**|**Ours**|**64.92**|
> ||25%|MoNE|33.94|
> ||25%, 4bit|He et al.|59.7|
> ||**25%, 4bit**|**Ours**|**67.61**|
> |DS-V2-Lite|0%|Baseline|68.6|
> ||50%|He et al.|53.88|
> ||50%|C-Prune|42.39|
> ||**50%**|**Ours**|**58.67**|
> ||25%, 4bit|He et al.|63.97|
> ||25%, 4bit|C-Prune|61.81|
> ||**25%, 4bit**|**Ours**|**67.61**|
>
> > Tab. R3
>
>
>
> **[Q2] Missing quantized baselines**
>
> P38%Q3.03b is not our standard setting. It is EAC-MoE’s best configuration.
>
> We quantize the baselines that are compatible with quantization in Tab. R4, with per-task results in Tab. R10.
>
> |Model|Setting|Method|Avg|
> |-|-|-|-|
> |Qwen1.5-MoE-A2.7B|0%|Baseline|68.32|
> ||25%,4bit|Wanda|68.02|
> ||25%,4bit|MoNE|62.17|
> ||25%,4bit|C-Prune|63.75|
> ||**25%,4bit**|**Ours**|**69.07**|
> |DS-V2-Lite|0%|Baseline|68.6|
> ||25%,4bit|Wanda|69.2|
> ||25%,4bit|MoNE|63.97|
> ||25%,4bit|C-Prune|61.81|
> ||**25%,4bit**|**Ours**|**67.61**|
>
> > Tab. R4
>
> Applying low-bit quantization causes a small performance drop. Our method remains competitive under the same NF4 quantization.
>
>
>
> **[Q3] Channel- vs. expert-level pruning**
>
> We add a comparison to expert-level pruning under matched pruning ratios:
>
> |Method|P25%|P38.3%|P50%|P61.7%|P75%|
> |-|-|-|-|-|-|
> |C-Prune|64.93|60.12|55.23|52.73|44.62|
> |MoNE|64.36|60.45|57.72|54.66|45.35|
> |He et al.|61.83|58.44|50.21|51.2|43.93|
> |**Ours**|**66.34**|**65.29**|**62.98**|**59.85**|**56.91**|
>
> > Tab. R5
>
> We also provide a Pareto frontier plot and per-task results in Fig. R1 and Tab. R11.
>
> Across all tested budgets, our method stays on the Pareto frontier and consistently outperforms expert-level pruning baselines, shows a stronger accuracy-compression tradeoff.
>
>
>
> **[Q4] A wider range of P/Q combinations**
>
> Based on our current experiments, P25%Q4b gives the best accuracy-efficiency tradeoff, but we do not claim it is a universal global optimum.
>
> We evaluate a broader range of P and Q, with per-task results in Tab. R12.
>
> |P%|Q|Storage (GB)|Avg|
> |-|-|-|-|
> |0%|16|28.7|62.98|
> |15%|8|13.5|61.47|
> |25%|8|12.2|60.57|
> |35%|8|11.0|60.28|
> |45%|8|9.8|60.23|
> |15%|4|7.4|60.34|
> |**25%**|**4**|**6.7**|**59.24**|
> |35%|4|6.1|59.65|
> |45%|4|5.4|56.84|
>
> > Tab. R6
>
> Stronger compression gives lower storage but larger accuracy drop, while milder compression preserves accuracy better. Different deployment scenarios may benefit from different trade-offs. Importantly, our framework allows for flexible trade-offs.

---

> > ### Author Rebuttal · Reviewer_fDBp · 2026-04-01
> >
> > I thank the authors for their thorough and constructive rebuttal. The responses have adequately addressed my previous concerns. The revisions are reasonable and improve the clarity and quality of the work. Accordingly, I have raised my rating to 5.

---

### Official Review · Reviewer_bCLh · 2026-03-10

**Soundness:** 3
**Presentation:** 3
**Significance:** 3
**Originality:** 3
**Overall Recommendation:** 4
**Confidence:** 3

**Summary:**

This paper proposes a structural pruning framework for MoEs: an attribution-guided loss approximation for efficient expert importance estimation, a coverage-maximized budget allocation that reframes pruning as maximizing channel score coverage under a global budget, and an alignment-aware redistribution for hardware-compatible low-bit inference.

**Compliance With Llm Reviewing Policy:**

Affirmed.

**Final Justification:**

Thanks to the authors for the feedback and justification, and to all the reviewers for the discussions. I am glad to support the positive rating.

**Key Questions For Authors:**

This is a solid, well-executed paper that addresses a practical, important problem in MoE compression. The proposed framework is principled, the results are good, and the engineering contributions are solid. Besides the weakness part above,

How sensitive is the method to the choice of calibration dataset?

**Limitations:**

yes

**Strengths And Weaknesses:**

**Strengths**

- The paper identifies a genuinely useful observation: channel-level saliency within MoE experts is highly concentrated, making expert-level importance too coarse for fine-grained pruning.

- Results on large-scale modern MoEs, including Qwen3, are convincing, and the attention to practical deployment details (kernel alignment, throughput measurement) reflects much engineering effort.

**Weaknesses**

- Some design choices, such as the square-root smoothing of layerwise loss, are presented without detailed justification. The paper would also benefit from a broader discussion of how the method generalizes to MoEs for visual tasks.

---

> ### Author Rebuttal · Authors · 2026-03-30
>
> We thank Reviewer bCLh for the positive feedback and constructive suggestions.
>
> Additional Fig. R1, Tab. R4, and references: [For-Reviewer-bCLh.md](https://anonymous.4open.science/r/moe-prune/additional/For-Reviewer-bCLh.md)
>
>
> **[W1] Square-root smoothing of layerwise loss lacks justification**
>
> We thank the reviewer for this suggestion.
>
> **Motivation.** The square-root smoothing is not a theoretically essential component. It is a simple way to compress the dynamic range of layerwise losses. Without smoothing, a few high-loss layers may take most of the channel budget, while low-loss layers may be over-pruned. We therefore apply a monotone concave transform to reduce this imbalance.
>
> **Why square root.** The square-root function is one such instance. The function $g(x) = \sqrt{x}$ for $x \geq 0$ is monotone (preserving relative layer ordering) and concave (suppressing outlier values), which makes the losses less sensitive to outliers. The key idea is **dynamic range compression**, not the square-root function itself.
>
> To clarify this, we further test several standard alternatives:
>
> (1) Log smoothing
>
> $$
> g(x)=\log(1+\alpha x), \quad \alpha>0
> $$
>
> (2)  Huber-style smoothing
>
> $g(x)=x$ if $x\le\delta$,
>
> $g(x)=\delta+\sqrt{\delta(x-\delta)}$ if $x>\delta$,
>
> where $\delta>0$ and $\delta=\mu+0.5\sigma$. $\mu$ and $\sigma$ are the mean and standard deviation.
>
> (3) Clip-based smoothing
> $$
> g(x)=\operatorname{clip}(x,\mu-k\sigma,\mu+k\sigma)
> $$
> where $k$ is the clipping cofficient.
>
> The smoothed loss curves are visualized in Fig. R1.
>
> We also compare downstream performance in Tab. R1, with details in Tab. R4.
>
> | Smooth fn       | Setting                | Avg       |
> | --------------- | ---------------------- | --------- |
> | NA              | -                      | 52.40     |
> | Log             | $\alpha=5$             | 56.92     |
> | Huber           | $\delta=\mu+0.5\sigma$ | 55.92     |
> | Clip            | $k=0.5$                | 56.69     |
> | **Sqrt (ours)** | **-**                  | **58.27** |
>
> > Tab. R1
>
> All smoothed variants outperform the unsmoothed baseline, which supports the need for dynamic range compression. The square-root transform is therefore not the only option, but a simple and effective one.
>
> **Revisions in paper:** We will add the motivation for smoothing, alternative smoothing results, and the visualization of smoothed loss curves.
>
>
>
> **[W2] Generalization to visual tasks**
>
> We agree that this is an important direction. Since our method operates on the MoE backbone and expert FFN layers, it is not limited to language-only MoEs and can also be applied to vision-language MoE models.
>
> To validate this, we additionally evaluate our method on Qwen3-VL-30B-A3B, a multimodal MoE model. The results are shown below.
>
> | Pruning ratio | OCRBench | MMBench |
> |---|---:|---:|
> | 0% | 66.5 | 87.0 |
> | 25% | 62.8 | 81.0 |
> | 50% | 44.6 | 72.0 |
>
> > Tab. R2
>
> These results provide initial evidence that our method can extend beyond language MoEs to visual tasks. Due to the limited rebuttal time, we will add broader evaluation on more vision tasks, multimodal MoE models, and baseline comparisons in our revision.
>
>
> **[Question 1] Sensitivity to calibration dataset**
>
> We appreciate the reviewer's constructive comments. Our original setup follows common practice in post-training compression: using C4 for general tasks, GSM8K for math, and OpenCodeReasoning for code. This is also consistent with prior work such as GPTQ [R1], DecoupleQ [R2], QuIP [R3], Arai et al. [R4].
>
> To study this more systematically, we conduct an additional ablation on six calibration corpora: WikiText2, C4, Pile[R5, R6], RedPajama [R7], GSM8K, and OpenCodeReasoning. Results are shown in Tab. R3.
>
> |Calibration Data|HellaSwag|Winogrande|PIQA|ARC-c|MMLU|GSM8K|HumanEval|
> |-|-|-|-|-|-|-|-|
> |WikiText2|57.83|**70.26**|74.85|36.11|51.59|27.15|2.44|
> |C4|**61.28**|69.26|74.45|35.94|52.46|9.18|3.05|
> |Pile|58.68|65.27|75.65|**39.92**|**54.18**|42.51|16.46|
> |GSM8K (train)|61.22|65.45|**76.05**|37.13|49.35|**61.08**|15.85|
> |OpenCodeReasoning|58.25|62.28|74.85|33.95|52.56|54.49|**29.27**|
> |RedPajama|60.61|65.88|73.65|36.51|52.43|23.35|3.05|
>
> > Tab. R3
>
> **Key findings:**
>
> 1. **General tasks are relatively robust** to general-domain calibration corpora such as WikiText2, C4, Pile, and RedPajama.
> 2. **Domain-specific tasks benefit from matched calibration.** GSM8K works best for math, and OpenCodeReasoning works best for code.
> 3. The sensitivity is **structured rather than arbitrary**. General corpora work well for general tasks, while domain-matched corpora work better for specialized tasks.
>
> **Revisions in paper:** We will add this calibration sensitivity ablation in the appendix and clarify the rationale for calibration data choice in the experimental setup.

---

> > ### Author Rebuttal · Reviewer_bCLh · 2026-04-04
> >
> > Thank the authors for the feedback. The feedback included a comparative table showing that, while multiple smoothing functions work, the square root performs best. The authors are still advised to discuss the sensitivity of the chosen hyperparameters.

---

> > > ### Author Response · Authors · 2026-04-07
> > >
> > > We thank the reviewers for their careful observation. We would like to clarify that we do not claim the square root transform is the universally best choice. As we said in our last reply, *all smoothed loss variants outperform the unsmoothed baseline, supporting the benefit of dynamic range compression. The square-root transform is therefore not the only option, but a simple and effective one.*
> > >
> > > To address the reviewer's concern, we conduct an additional parameter sweep for different smoothing functions, and the results are shown in the table below. It suggests that (1) the specific hyperparameters does affect the performance, and the smoothed losses consistently outperforms the raw loss baseline, (2) other smoothing functions can achieve very similar behavior as ours. For example, as shown in [Fig. R1](https://anonymous.4open.science/r/moe-prune/additional/For-Reviewer-bCLh.md), the log function with $\alpha = 20$ produces a smoothing curve that is nearly overlapping with our square-root variant, and its downstream results are also highly comparable to ours.
> > >
> > > Due to the limited rebuttal period, we are not able to exhaustively explore a larger hyperparameter space. We agree that the choice of smoothing function and the hyperparameter are important, and we will add a more explicit discussion with additional parameter sweeps results in the revised paper.
> > >
> > > | Smooth fn | setting | PIQA | ARC-c | ARC-e | BoolQ | HellaSwag | WinoGrande | Avg |
> > > | - | - | - | - | - | - | - | - | - |
> > > | raw| - | 72.06 | 34.73| 54.29| 70.46 | 58.48| 64.07 | 59.02 |
> > > | log| $\alpha=5$| 74.85 | 32.34| 60.88| 68.66 | 61.07| 68.86 | 61.11 |
> > > | log| $\alpha=10$ | 76.13 | 36.53| 64.74| 70.71 | 61.84| 68.72 | 63.11 |
> > > | log| $\alpha=20$ | 76.13 | 37.25| 64.74| 71.43 | 61.12| 70.16 | 63.47 |
> > > | clip | k=0.25 | 75.95 | 36.71| 63.65| 70.16 | 61.84| 68.35 | 62.78 |
> > > | clip | k=0.5| 74.45 | 35.33| 59.88| 68.66 | 62.08| 68.66 | 61.51 |
> > > | clip | k=1.0| 74.14 | 35.08| 59.86| 68.17 | 61.30| 68.17 | 61.12 |
> > > | Huber| $\delta=\mu+0.25\sigma$ | 74.14 | 33.45| 57.32| 67.81 | 61.48| 67.99 | 60.37 |
> > > | Huber| $\delta=\mu+0.5\sigma$| 73.45 | 32.14| 56.89| 67.66 | 61.65| 69.86 | 60.28 |
> > > | Huber| $\delta=\mu+1.0\sigma$| 75.05 | 34.72| 59.31| 66.91 | 61.30| 66.73 | 60.67 |
> > > | **Power-law (ours, sqrt)** | $p=\frac{1}{2}$| 74.45 | 35.93| 66.07| 70.46 | 61.28| 69.26 | 62.91 |
> > > | **Power-law**| $p=\frac{1}{3}$| 75.77 | 38.16| 66.73| 69.62 | 61.66| 68.90 | 63.47 |
> > > | **Power-law**| $p=\frac{1}{4}$| 75.59 | 38.52| 67.27| 69.80 | 62.03| 69.26 | 63.75 |
> > >
> > > > Power-law: $\mathcal L_\mathrm{smooth}=\mathcal L_\mathrm{raw}^p$. $p=\frac{1}{2}$ is the square-root transform used in our main results, $p=\frac{1}{3}$ and $p=\frac{1}{4}$ correspond to cube-root and fourth-root.

---

### Official Review · Reviewer_qUVM · 2026-03-13

**Soundness:** 3
**Presentation:** 3
**Significance:** 2
**Originality:** 2
**Overall Recommendation:** 4
**Confidence:** 4

**Summary:**

The authors propose a channel level structural pruning method for MoEs with observations that useful information within experts is concentrated in a small number of channels. Based on this, they use attribution based importance scoring and a coverage maximization objective to select channels to keep while pruning redundant ones. Experiments on DeepSeek and Qwen MoE models show substantial memory reduction, especially when combined with 4-bit quantization, while maintaining competitive performance.

**Compliance With Llm Reviewing Policy:**

Affirmed.

**Key Questions For Authors:**

In particular, the paper would benefit from additional analysis such as:

- comparisons at similar compression levels where competing methods may achieve higher performance, or
- comparisons at similar performance levels where the proposed method achieves stronger compression.

Specifically, if the authors are able to provide pareto frontier plots of performance vs. compression, it would help clarify the tradeoffs.

**Limitations:**

The discussion of limitations and broader impacts is relatively limited. The paper could be strengthened by addressing several limitation discussions:
1. the effectiveness of the approach may depend on the routing behavior and expert activation distributions of specific MoE architectures, which could vary across models or tasks, it would be helpful to clarify how robust the method is under different routing dynamics or expert heterogeneity
2. the pruning strategy relies on first order approximations, which may introduce estimation error and potentially affect pruning decisions in cases where higher order interactions between channels are important.

**Strengths And Weaknesses:**

Strengths

- Soundness: The authors begin with observation from past methods: most existing MoE pruning methods operate at the expert level, which can be overly coarse and may overlook redundancy within individual experts. With this insight, the paper proposes a structured pruning framework at the channel level. The method further introduces a coverage based objective, combined with attribution guided importance estimation, to guide how pruning budgets are distributed across experts and layers. The formulation and the algorithm design aligns well with the motivation presented earlier in the paper. Experiments are conducted on several large MoE models, including DeepSeek and Qwen, and across a range of pruning ratios. The evaluation further considers realistic deployment settings by combining pruning with quantization, pushing further to aggressive compression ratio.

- Presentation: The motivation is presented clearly: the authors first point out the limitations of expert level pruning and then highlight redundancy that exists inside experts. This naturally motivates the move to channel level pruning. The experimental section is comprehensive and compares against several existing baselines, which helps place the proposed method within the broader literature.

- Significance: Reducing memory usage and inference cost of MoEs is critical for real-world deployment. This paper tackles this problem by introducing a structured pruning.

- Originality: The paper offers an interesting perspective on MoE pruning by shifting the focus from pruning entire experts to identifying redundancy at the channel level within experts. While some components, such as attribution based importance estimation and structured pruning, are related to prior work, their combination in the context of MoE channel pruning is reasonably novel and supported by empirical observations.




Weaknesses

- Presentation: Sections 4.2 and 4.3 describing the methodology are somewhat verbose and could benefit from clearer and more concise writeup. The sections contain several repeated explanations and claims that make the core algorithm harder to follow. Streamlining these sections and focusing on the essential algorithmic steps would improve readability and reduce potential confusion for readers.

- Soundness / Experimental Analysis: Although the experimental section is comprehensive and evaluates many baselines, the performance results do not always appear to dominate competing methods. especially the accuracy or downstream performance are not consistently the best among compared methods. It would strengthen the paper to more clearly contextualize these tradeoffs.

- Originality: The novelty components of the method are attribution based importance scoring, and coverage-based objectives. Theseare  built on ideas that have appeared in prior compression and pruning literature. The novelty mainly arises from their integration and application to MoE channel pruning.

---

> ### Author Rebuttal · Authors · 2026-03-30
>
> We thank Reviewer qUVM for the careful review and constructive suggestions.
>
> Additional Fig. R1-R3, Tab. R5-R8, definition of scores, and references: [For-Reviewer-qUVM.md](https://anonymous.4open.science/r/moe-prune/additional/For-Reviewer-qUVM.md)
>
>
>
> **[W1] Sections 4.2 and 4.3 are verbose**
>
> We will shorten Secs 4.2 and 4.3 by removing repeated motivation and non-essential discussion.
>
>
>
> **[W2] Performance does not consistently dominate**
>
> We agree it should be stated more clearly.
>
> 1. Our method uses structural pruning and 4-bit quantization, and is directly deployable. Real storage and memory savings, and throughput are shown in Fig. 6 and Fig. 8. Competing methods may face deployment barriers that is not natively supported on commodity hardware, such as Wanda, EAC-MoE, and MoE-I$^2$.
> 2. Under the same compression ratio, our method outperforms other structural pruning methods (performance-compression Pareto frontier can be found in **[Q1]**).
>
>
>
> **[W3] Novelty**
>
> Some components are inspired by prior work, but our contribution is not a simple combination:
>
> 1. Coverage-Maximized Budget Allocation (CBA) have fundamentally different concept of *Coverage* from prior methods such as REPrune[1]. REPrune maximizes representative coverage in clustered CNN kernels, while our CBA maximizes the retained channel scores globally.
>
> 2. We are the first to adapt attribution-based scoring to MoE expert heterogeneity, and introduce Hamilton redistribution for hardware alignment constraints.
>
> 3.  Key novelty also lies in the unified pipeline, which makes aggressive compression practical while preserving deployability.
>
>
>
> **[Q1] Pareto Frontier plots**
>
> We will add a **performance-compression Pareto frontier** to make these tradeoffs explicit.
>
> |Method|P25%|P38%|P50%|P62%|P75%|
> |-|-|-|-|-|-|
> |C-Prune|64.93|60.12|55.23|52.73|44.62|
> |MoNE|64.36|60.45|57.72|54.66|45.35|
> |He et al.|61.83|58.44|50.21|51.2|43.93|
> |**Ours**|**66.34**|**65.29**|**62.98**|**59.85**|**56.91**|
>
> > Tab. R1
>
> More results and Pareto frontier plots are in Fig. R1 and Tab. R5. Across all tested budgets, our method consistently outperforms expert-level pruning baselines. This shows that channel-level pruning gives a better accuracy-compression tradeoff.
>
>
>
> **[L1] Robustness under routing dynamics or expert heterogeneity**
>
> Our method is not tied to specific router design or activation pattern.
>
> **Router dynamics.** The current experiments already cover Qwen-style standard top-k routing, and more constrained routing of DeepSeek with load balancing considerations. We visualize the router entropy statistics for Qwen and DS MoEs in Fig. R2 to show the dynamics.
>
> **Top-1 / Top-2 routing.** Results are listed in Tab. R2, with per-task results in Tab. R6. Accuracy drops when fewer experts are activated as expected, but our method still preserves most of the original performance under all routing policies.
>
> |Model|Top-k|P%|Avg|
> |-|-|-|-|
> |Qwen1.5-MoE-A2.7B|2|0%|62.98|
> ||2|50%|58.27|
> ||1|0%|60.56|
> ||1|50%| 55.15|
> |DS-V2-Lite|2|0%|54.84|
> ||2|50%|50.12|
> ||1|0%|62.23|
> ||1|50%|59.08|
>
> > Tab. R2
>
> **Expert heterogeneity.** Activation visualizations are shown in Fig. R3, confirming expert heterogeneity across tasks and layers. Even so, our method remains consistently effective, as shown in Tab.s 3, 4, and 5 of the main paper.
>
>
>
> **[L2] First-order approximation may introduce estimation error**
>
> In the original submission, we use the first-order because it is efficient and widely used, while exact second-order computation on full weight matrices is too expensive [R1, R2, R3]. To address this, we introduce a **lightweight but exact second-order proxy**. By introducing a scalar perturbation $\alpha_e$ for each expert output, we have
>
> - $-\frac{\partial \mathcal L_e}{\partial \alpha_e}$: first-order removal gain ($s_e^{(1)}$)
> - $-\frac{\partial \mathcal L_e}{\partial \alpha_e} + \frac{1}{2}\frac{\partial ^2 \mathcal L_e}{\partial \alpha^2_e}$: second-order removal gain ($s_e^{(2)}$)
>
> We compare the three scores in Tab. R3. $s_e^{(\mathrm{true})}$ is the true local perturbation loss.
>
> |Comparison|Pearson ↑|L1 ↓|Channel Pearson ↑|Channel Diff ↓|
> |-|-|-|-|-|
> |$s_e^{(1)}$ vs $s_e^{(2)}$|0.959|0.058|0.966|23.4|
> |$s_e^{(1)}$ vs $s_e^{(\mathrm{true})}$|0.959|0.058|0.966|23.4|
> |$s_e^{(2)}$ vs $s_e^{(\mathrm{true})}$|1.000|0.000|1.000|0.13|
>
> > Tab. R3
>
> Results show that the first-order scrore is highly correlated with the second-order, and the resulting channel allocation is also very close.
>
> Runtime is reported in Tab. R7. The second-order costs **16.9x** more time than first-order.
>
> The end-to-end downstream performance is shown in Tab. R4, with detailed in Tab. R8.
>
> |Metric|Avg|
> |-|-|
> |$s_e^{(1)}$|60.89|
> |$s_e^{(2)}$|62.09|
>
> > Tab. R4
>
> The second-order proxy improves performance. We will include this as an enhanced variant in the revision.

---

> > ### Author Rebuttal · Reviewer_qUVM · 2026-04-01
> >
> > The rebuttal meaningfully strengthens the paper, particularly with the added Pareto analysis and second order validation, these really help strengthen the results and address my concerns.
> > Regarding novelty, I maintain my view that the work represents a non trivial and well motivated reformulation of pruning for MoE models, combining known components into a coherent and practically impactful framework. Overall, I keep my positive rating.

---

### Decision · Program_Chairs · 2026-04-30

**Decision:**

Accept (spotlight)

**Comment:**

The paper solves a critical problem - that is hardware aware compression of MoE networks with channel level compression. All the reviewers agree that the problem is indeed strong, the paper is very well-written and the techniques involving attribution guided loss approximation through first-order taylor approximation, novel methods for coverage computation and hardware-based redistribution make this a strong contribution.

Some reviewers raised concerns regard design choices (square root smoothing, first order loss approximation) - the detailed rebuttals from the authors assuaged them entirely. The rebuttals in this case are strong, detailed and contain a wealth of information - in particular, I also liked the pareto frontier performance-compression results. I strongly suggest the authors to include these results in the final version of the paper.